# Bio-Tailored Sensing at the Nanoscale: Biochemical Aspects and Applications

**DOI:** 10.3390/s23020949

**Published:** 2023-01-13

**Authors:** Francesca Fata, Federica Gabriele, Francesco Angelucci, Rodolfo Ippoliti, Luana Di Leandro, Francesco Giansanti, Matteo Ardini

**Affiliations:** Department of Life, Health and Environmental Sciences, University of L’Aquila, 67100 L’Aquila, Italy

**Keywords:** biomolecules, nanomaterials, nanostructures, label-free detection, biorecognition, proteins, aptamers, electrochemistry, optics, bioconjugation

## Abstract

The demonstration of the first enzyme-based electrode to detect glucose, published in 1967 by S. J. Updike and G. P. Hicks, kicked off huge efforts in building sensors where biomolecules are exploited as native or modified to achieve new or improved sensing performances. In this growing area, bionanotechnology has become prominent in demonstrating how nanomaterials can be tailored into responsive nanostructures using biomolecules and integrated into sensors to detect different analytes, e.g., biomarkers, antibiotics, toxins and organic compounds as well as whole cells and microorganisms with very high sensitivity. Accounting for the natural affinity between biomolecules and almost every type of nanomaterials and taking advantage of well-known crosslinking strategies to stabilize the resulting hybrid nanostructures, biosensors with broad applications and with unprecedented low detection limits have been realized. This review depicts a comprehensive collection of the most recent biochemical and biophysical strategies for building hybrid devices based on bioconjugated nanomaterials and their applications in label-free detection for diagnostics, food and environmental analysis.

## 1. Introduction

Since 1967, when S. J. Updike and G. P. Hicks showed the first enzyme-based electrode for glucose detection [1] inspired by the previous L. C. Clark Jr.’s device [2], sensors for biological analytes or “biosensors” have passed through a three-generations development [3]. In those belonging to the third generation group the sensing molecule is physically integrated within the device rather than being freely diffusing in solution. This setup resulted in improved detection as it does not require a mediator component to transfer the signal to the transducer thereby becoming simpler and faster.

Modern biosensors are usually endowed with (i) efficient capturing of specific targets with no off-target interaction, (ii) transformation into electrochemical, electrical, optical, gravimetric or acoustic signals, (iii) sensitivity, short response time, reproducibility and low detection limit (LOD) and (iv) miniaturized components [3]. In the attempt to improve these features, great results have been achieved by taking advantage of 0D, 1D and 2D nanomaterials for their unique chemical and physical properties, which make them highly performing compared to the bulk counterparts. Biomolecules represent naturally occurring ready-to-use nanomaterials as they are nanoscopic objects with regular shape, high binding specificity and affinity, surfaces suitable for chemical functionalization and catalytic activity as in the case of enzymes, for instance [4]. Moreover, biomolecules can be modelled via genetic and protein engineering [5,6] and their surfaces enable the interaction with other nanomaterials; as a result, biomolecules have been widely exploited to functionalize nanomaterials thus enabling precise tailoring of hybrid functional bioconjugates at the nanoscale [7,8,9]. Beyond biomolecules, organic and inorganic nanomaterials with sharpen architecture have attracted interest for their high surface-to-volume ratio, conductivity, shock-bearing ability and optical tunability, e.g., nanoparticles, nanotubes, nanorods, graphene, nanofibers, molybdenum disulfide, carbides, nitrides, carbonitrides and quantum dots [3]. Their features have impelled biosensing accuracy and robustness and disclosed tunable electrochemical and physical-mechanical responses. For instance, nanomaterials integrated in a sensor can sensibly decrease the LOD and thus increase the sensitivity in minimal processing time to reach high specific detection of a target [10,11,12].

To note, as the sensors get miniaturized to the nanoscale, the so-called “label-free” detection of molecular analytes becomes affordable. One first advantage of using labeled-free detection, indeed, is the possibility of revealing molecular targets which are not easy to tag or cannot be tagged at all; secondly, many targets cannot be labeled for they are found at very low amounts thus making labeling expensive and a nonsense strategy. In such a scenario, the interplay between biomolecules and nanomaterials becomes straightforward because at the nanoscale the interactions with the target analytes turn highly specific, strong and favored; moreover, these interactions become affected by other factors which are absent in bulk macroscopic devices, for instance the molecular conformation of the biomolecules. Under these conditions, the detection inspired by structure-based high affinity biorecognition turns on and accurate targeting of the analyte can be addressed with no off-target issues even at very low concentrations.

The past two decades gave us pretty examples of biosensors based on nanometric components and highlighted the advantages of having a biological element as the sensing part [13]. Mentioning all these examples would be an endless list for such a flourishing and interdisciplinary applied science; however, a few representative examples of biosensors might be quickly recalled, e.g., DNA-coated nanoparticles for enzymatic activity detection [14], lectin-coated nanorods for detection of bacterial cells [15], antibody-coated nanowires for specific protein detection [16] and antibody-coated micropillars with nanometric roughness to detect cancer cells [17]. Beyond the ease of combining a biological moiety, be it an enzyme, nucleic acid or a protein, with a non-biological nanomaterial by means of simple adsorption or chemical crosslinking, the main advantage of obtaining such a hybrid conjugate is a reduced LOD and, therefore, increased sensitivity coming from the intrinsic properties of both the biomolecule and the nanomaterial [13,18]. To get an idea, a method based on silicon functionalization and gold nanoparticles covalently attached to antibodies enable targeting of the prostate-specific antigen within a working range of 23 fg mL^−1^–500 ng mL^−1^ with an LOD of 23 fg mL^−1^, which corresponds to target concentrations in the femto range (10^−15^ M) [19]. Thus, it is now clear that integrating an active biological moiety onto an organic or inorganic nanometric substrate is propaedeutic to realize advanced, next-generation devices for improved sensing.

In this review, several recent examples published since 2020 of label-free biosensors based on biomolecules and nanomaterials and their applications are collected and discussed. The last two years, indeed, have been propelled by many different devices where aptamers, proteins, enzymes and/or antibodies have been found as the most common biological sensing components. Note that in this review the terms “nanomaterials” and “nanostructures” exclusively refer to nanometric objects having a regular, sharpen architecture, e.g., 0D particles, dots and rods; 1D tubes and fibers; 2D and 3D organic and inorganic layered-, porous- or fiber-like matrices. Also note that, due to the huge area of label-free techniques [3,20], this review will mainly focus on biosensors exploiting electrochemical or optical features for being the most representative (Figure 1). A special emphasis is given to the biochemical and biophysical aspects concerning the strategy to assemble the hybrid bioconjugate, the chemical and/or physical signals detected and the applications ranging from detection of biomarkers and antibiotics, environment- and food-derived molecules and single-molecule detection.

## 2. Biosensors Based on Hybrid Biomolecules-Nanomaterials Composites

Each example, discussed in this section, is provided with a description of the chemical and/or biochemical strategies to couple the sensing biomolecular component to the nanomaterial or nanostructure and its application for practical purposes. A series of examples will also be quickly recalled that rely on the same principle of the main example just described. As mentioned before, label-free biosensors involving nanomaterials or nanostructures with regular, sharpened architecture will be exclusively considered. Furthermore, this section will mainly provide examples of aptamers-, proteins-, antibodies- and enzymes-based biosensing due to their predominant abundance in the field.

### 2.1. Electrochemical Biosensors

#### 2.1.1. Aptamers-Based Electrochemical Biosensors

Aptamers are short, single-stranded nucleotide sequences or peptides that spontaneously adopt secondary conformations in solution under particular conditions, e.g., hairpin-like (stem loop) structures [21]. They are considered alternatives to antibodies as they are endowed with high binding affinity and specificity to various biological targets with little or no off-target interaction; furthermore, they show no immunogenicity, large molecular flexibility as well as thermal and chemical stability. Despite their natural occurrence, aptamers can be easily synthesized at low cost and are particularly targeted for different applications, including sensing. Indeed, though aptamers are well-known as very proficient therapeutic tools, for instance in cancer therapy [21,22,23], their application in biosensing has recently been propelled by many examples reporting on molecular, ion and cell detection in food and water as well as in diagnostics [24]. The combination of aptamers with nanomaterials is not an exception.

Most biosensors where aptamers and nanomaterials are physically integrated, so-called “aptasensors”, rely on an electrochemical setup that includes an electrode working as transducer and support for deposition of the nanomaterial, which improves the electrochemical properties of the underlying electrode and furnishes binding sites for the aptamers. In this configuration, the nanomaterial-bound aptamers act as receptors/probes that are freely accessible to capture and detect the target analytes. Namely, upon binding to the target the aptamers undergo a complementary base pairing that creates secondary structures, e.g., helical arms and single stranded loops, which in turn provide the information to form tertiary structures responsible for the high-specific and high-affinity binding to the target via weak forces. According to this structural plasticity, aptasensors show electrochemical performances that can be monitored via voltametric apparatus as, upon binding to the analytes, the molecular conformational change of aptamers alters the free surface of the electrode to accomplish an electron exchange, e.g., [Fe(CN)_6_]^3−/4-^, thus leading to an increased or decreased rate of redox reactions on the electrode and therefore to changes in the voltametric and current signals. It must be noted that the signal changes that occur during the interaction between the aptamers and the sample containing the target molecules is very dependent upon the time, which usually spans within tens of minutes before reaching a stable signal. A main drawback that must be avoided comes from the non-specific binding of the target on the surface of the electrode via adsorption that leads to an increase in the background signal therefore decreasing sensitivity of the device. In all cases, this drawback is usually fixed by allowing blocking agent to adsorb over the electrode surface, e.g., surfactants, metal-specific thionic compounds as well as proteins such as casein and bovine serum albumin (BSA). This expedient allows the target molecules to interact exclusively with the aptamer probes.

A representative example is reported by Beitollahi et al. [25] that developed an electrochemical aptasensor to detect in blood serum and urine the amino acid homocysteine, a highly reactive sulfur-containing compound that turns into an intermediate by-product during the metabolism of dietary methionine (Figure 2). In this study, gold nanoparticles (AuNPs) are synthesized via electrodeposition onto the surface of a glassy carbon electrode and exploited to stably bind thiolated aptamers (SH-aptamers) targeting homocysteine. The interaction between the AuNPs and the SH-aptamers occurs through involvement of strong thiolate-gold bonds, which allow the aptamer molecules to form a self-assembled monolayer on the metal surface, a molecular arrangement that is preferable to achieve linear detection of the analyte. The authors used voltametric measurements to reveal the binding of the homocysteine molecules to the aptamer probes with an LOD down to 0.01 μM and a linear response within the range 0.05–20.0 μM; indeed, the interaction between the aptamer probes and the homocysteine molecules triggers a fold change in the aptamer molecular structure that increases the accessibility of the underlying glassy carbon electrode leading to enhanced redox reactions and therefore increases the voltametric and current values. This aptasensor has been proposed as a potential device for healthcare purposes, especially for detecting homocysteine as a biomarker in hyperhomocysteinemia.

The aptasensor just described includes only three main components, i.e., the electrode (the transducer), a layer of NPs (the nanostructured support) and the aptamer (the biological probe). Likewise, the same simple though efficient configuration can be found in other devices having different components such as screen-printed gold electrodes coated with flower-like gold microstructures for detection of serpin in plasma [26] or coated with CeO_2_NPs to detect the epithelial sodium channel protein in urine [27].

As the complexity of the nanomaterial laying onto the electrode is increased or several nanomaterials are combined to produce more complex composites, the performances of the aptasensor are enhanced to reach very low LODs. For instance, the study by Xie et al. [28] is worth mentioning as it describes an aptasensor able to detect very low amounts of thrombin in blood serum with no interference from nonspecific adsorption, environmental changes or instrumental efficiency by taking advantage of a hybrid configuration (Figure 3). To realize this aptasensor, AuNPs are synthesized via heating in solution and are loaded onto a mixture reaction of aminated ionic liquid NH_2_-IL and 2D nanosheets of molybdenum disulfide (MoS_2_), the former helping the nanomaterial to stabilize via electrostatic repulsion and avoiding aggregation. The resulting AuNPs-IL-MoS_2_ composite is dried onto a glassy carbon electrode and furnishes a proper binding surface for the adsorption of DNA nanotetrahedrons, which are obtained by warming and cooling of four different SH-DNA sequences. The thrombin analyte molecules can bind to the aptamers and the resulting thrombin-DNA nanotetrahedron complex is sandwiched using a third nanostructured composite AuNPs-Fe-MIL-88 where AuNPs are loaded onto hydrothermally synthesized Fe-MIL-88 metal-organic frameworks (MOFs), which in turn are labeled with a fifth DNA sequence recognizing the thrombin-DNA complex. The authors recalled two import features of such high-complex devices: first, in contrast to traditional stem-loop structures or linear DNA probes, the enhanced mechanical rigidity of the DNA nanotetrahedrons increases the accessibility of the target molecules and their loading amounts; secondly, this aptasensor results bind to many interfering substances even when their concentrations are greater than the concentration of thrombin. This aptamersensor shows an LOD of 56 fM within a large linear range of 0.298–29.8 pM thus representing a potential detector for ratiometric methods in clinical applications.

This aptasensor represents a prominent example of how nanomaterials, according to their intrinsic features such as hydrophilicity, high specific surface area and good electrical conductivity, can sensibly improve the electrochemical performances of detection without affecting the biochemical features of the attached aptamers. These properties reflect the high number of similar devices which have been realized in recent years. Additional examples where the glassy carbon electrode is coated with hybrid nanostructured composites include: reduced MoS_2_-AuNPs for the double detection of zearalenone and fumonisin B1 mycotoxins in food [29]; (3-aminopropyl)triethoxysilane-modified graphene oxide (GO)-AgNPs for detection of chloramphenicol in food [30]; thiourea capped-ZnS quantum dots (QDs)-AuNPs for detection of β-casomorphin 7 in urine [31]; AgNPs-GQDs containing core-shell Cu-In-S/ZnSQDs for detection of ractopamine in urine and serum [32]; magnetic reduced GO-Fe_3_O_4_-Cu_2_O and Ag-resorcinol-formaldehyde NPs-Ag nanodots (NDs) for detection of prostate specific antigen in serum [33]; reduced GO-AgNPs and prussian blue-AuNPs for detection of acetamipridin in food [34]; reduced GO-AuNPs for detection of glycated albumin [35]; MXene-AuNPs for detection of chloramphenicol in food [36]; PtNPs-MIL-101(Fe) MOFs for detection of aflatoxin M1 in food [37]; and bimetallic CoNi-MOFs with graphene-like nanosheet structure to detect enrofloxacin in food, water and serum [38].

In this context, another prominent example is reported by Li et al. for detecting carcinoembryonic antigen in blood serum [39]. The aptasensor enables dual binding sites and dual signal amplifying electrochemical sensing by means of self-polymerized dopamine-functionalized AuNPs (Figure 4). AuNPs are chemically synthesized and decorated with dopamine moieties under oxidative and alkaline conditions to trigger dopamine polymerization over the gold surface. These polymeric hybrid constructs provide anchoring sites for Fe^3+^ through coordination bonds to form Fe^3+^-catechol MOFs with an octahedron shape and the presence of acetic acid increases the number of carboxyl groups (-COOH) regulating the nucleation and growth rate of the Fe^3+^-catechol MOFs on the surface of the AuNPs. The presence of several -COOH groups within the framework complexes is exploited to link carcinoembryonic antigen-specific aminated aptamers (NH_2_-aptamers) via classical 1-Ethyl-3-(3-dimethylaminopropyl)carbodiimide/N-hydroxysuccinimide (EDS/NHS) crosslinking reaction to achieve a colloidal aptasensor with high sensitivity and selectivity, good biocompatibility and strong stability. These functionalized AuNPs can be used to coat a glassy carbon electrode and detect carcinoembryonic antigen in serum. Note that both the polydopmine moieties and the Fe^3+^-catechol MOFs accelerate the electron transfer on the electrode interface, increasing the amplification of the current signals and therefore enhancing the sensitivity of the device. The authors used such a complex aptasensor to detect the carcinoembryonic antigen in the concentration range of 1 fg mL^−1^–1 μg mL^−1^ with a LOD as low as 0.33 fg mL^−1^ and proposed it for detection of the carcinoembryonic antigen biomarker in early clinical diagnosis.

The aforementioned examples can be further extended if considering other types of electrodes, such as: screen-printed carbon electrodes coated with carbon-black NPs-AuNPs composites to detect Cd(II) ions in water [40] or coated with exonuclease-assisted AuNPs-MOFs for detection of streptomycin in food [41]; fluorine-doped tin oxide electrodes coated with BiVO_4_-2D-C_3_N_4_ photoanode with II type heterojunction to detect microcystin-LR in water [42]; Au electrodes coated with highly-porous Au nanostructures for detection of acetamiprid in food [43]; indium tin oxide electrodes coated with core-shell LaFeO_3_-g-C_3_N_4_ heterostructures for detection of streptomycin in food [44]; Au microelectrodes coated with GQDs-AuNPs composites for detection of vitamin D in serum [45]; screen-printed graphite electrodes coated with poly-3-amino-1,2,4-triazole-5-thiol-GO-AuNPs composites for detection of lipocalin-2 in serum [46]; indium tin oxide electrodes coated with TiO_2_-BiOI-BiOBr visible-light photosensitive material to detect streptomycin in food and serum [47]; screen-printed carbon electrodes coated with magnetic GO-Fe_3_O_4_ to detect organophosphorus pesticides in food [48]; Pt electrodes coated with AgNPs and reacted with luminol-hydrogen peroxide to detect kanamycin in food [49]; Au electrodes coated with polyethyleneimine-functionalized reduced GO-AuNCs for detection of chloramphenicol in food [50]; Au electrodes coated with polyethyleneimine-graphite-like carbon nitride/Au nanowires (AuNWs) for detection of chloramphenicol in food [51]; Au electrodes coated with Au nanospheres (AuNSs) and functionalized with CdSQDs and PdSQDs for detection of kanamycin and tobramycin in serum [52]; and exonuclease-assisted Au electrodes coated with DNA nanostructures and horseradish peroxidase-functionalized AuNPs for detection of kanamycin in food [53].

In this context, an example of an aptasensor developed by Lv et al. [54] is significative for involving unusual electrode-supported dendrite-like nanostructures. In this example, carbon cloth-supported Au nanodendrites functionalized with AuNPs and terminal-free aptamers are realized for sensing sialic acid in blood serum (Figure 5). To realize the device, large 1 × 1.5 cm carbon cloth electrodes are used as substrate for synthesis via the electrodeposition of Au nanodendrites protruding to the solvent solution to guarantee high surface area and high conductivity along the carbon length. The resulting nanodendrites can be coated via simple adsorption with unfunctionalized aptamers targeting sialic acid. Colloidal AuNPs are then synthesized via a chemical reduction and functionalized with 4-mercaptophenylboric acid and thionine via boronic acid-diol binding. The aptamers are exploited as the sensing element from blood followed by the attachment of the modified AuNPs where the electrochemical reduction of the thionine moieties triggers the signal amplification all along the entire large surface of the gold nanodendrites. The device has been described to detect sialic acid concentrations up to LOD of 60 nM within a linear range of 0.1–440 μM and proposed to evaluate sialic acid levels in human serum.

Table 1 provides a list of the current electrochemical aptasensors reported in this review and their main features.

#### 2.1.2. Proteins-Based Electrochemical Biosensors

Amongst biomolecules, proteins represent the larger and most intriguing group for their multiple biochemical, structural and functional features. Briefly, proteins, thanks to their self-recognition and self-assembling properties, can form macromolecular 3D globular or filamentous structures ranging from a few to millions in Daltons molecular weight [4]. Solving, predicting or creating ex novo protein structures is a challenging problem in bioinformatics but of enormous scientific interest in fundamental and applied sciences from biotechnology to medicine [55]. Unfortunately, unlike aptamers, most of the proteins are chemically and thermally unstable; furthermore, de novo synthesis with desired characteristics is challenging though many advanced techniques allow routine production from lab-scale to industrial scale. However, these drawbacks come with features which place proteins at the forefront of biotechnological and bionanotechnological applications, e.g., self-assembly behavior to form sharpened complexes, high binding affinity and specificity with other biological targets and pronounced enzymatic activity. In addition, proteins are the best biological tools to build advanced, hybrid and responsive nanostructures [56,57,58,59,60,61].

Like aptamers, proteins have been widely exploited as probes in sensing applications where traditional voltametric electrode-based sensors are the most explored to reveal the presence of specific analytes. Antibodies are found in most of the biosensors reported in literature, which are referred to as “immunosensors”. The basic principle of immunosensors recalls the one exploited for aptasensors as it is based on a high specificity and high affinity binding to the target analyte that affects the electrochemical reaction rate on the electrode surface. Accordingly, coating the surface of the electrode with a blocking agent is one main condition to accomplish and to avoid non-specific adsorption and background noise. An example is shown by Hartati et al. [62] that realized an electrochemical immunosensor based on antibodies-conjugated AuNPs to detect the epithelial sodium channel, a protein considered a biomarker in hypertension (Figure 6). The AuNPs are prepared using a wet synthesis through chemical reduction and reacted with SH-polyethylene glycol-COOH molecules through the formation of thiolate-gold bonds in order to coat the surface of the gold with –COOH monolayers. In this configuration, the surface of the AuNPs can be functionalized with antibodies specific for the epithelial sodium-channel protein acting as a molecular probe for high affinity and high-specificity recognition via the crosslinking reagents EDC/NHS that forms stable covalent bonds between the COOH– functional groups and the native NH_2_– groups located on the antibody surface. In these conditions, the polyethylene glycol molecules serve as spacers between the metal surface and the antibody molecules thus preventing nonspecific adsorption. As a voltametric apparatus, Au-screen-printed carbon electrodes are obtained through the electrodeposition of Au onto the surface of the carbon electrode followed by a coating with cysteamine, the latter acting as linkers between AuNPs and antibodies. The resulting modified electrode is used to detect via voltametric measurements the epithelial sodium-channel protein molecules in urine samples with an LOD of 2.8 × 10^−1^ ng mL^−1^ in the range of 9.375 × 10^−2^–1.0 ng mL^−1^ and is proposed as a biosensor for suspected hypertensive urine sample patients and an alternative method to replace the ELISA screening.

This immunosensor involves the most common and simple configuration based on NPs to detect biomolecules and it is representative of many other similar studies such as those reporting on CeO_2_NPs-antibodies on screen-printed carbon, as follows: AuNPs electrodes for detection of Herceptin-2 in serum [63], divalent metal ions-functionalized AuNPs-carbon NSs-antibodies on glassy carbon electrodes for simultaneous detection of prostate specific antigen, carcinoembryonic antigen and α-fetoprotein in serum [64], magnetic Fe_3_O_4_NPs-antibodies on Au electrodes to detect Siglec 15 protein in serum [65], poly(2-hydroxyethyl methacrylate-glycidyl methacrylate)NPs-ureases on screen-printed carbon electrodes for urea detection in serum [66] and AgNPs-antibodies on graphite electrodes to detect tick-borne encephalitis virus in serum [67]. Other nanostructures can be recalled herein such as those reporting on SnS_2_ nanoflake-chitosan composites functionalized with biotinylated antibodies for the detection of carcinoembryonic antigen in serum [68].

When approaching the use of more advanced nanomaterials in building efficient biosensors, proteins, enzymes and antibodies are largely considered as their intrinsic biochemical and structural features make them ideal candidates to assemble hybrid composites while keeping useful properties such as enzymatic activities, self-assembly behavior and biorecognition. As a first example, Chanarsa et al. [69] reported a AgNPs-reduced GO composite as a component of an electrochemical immunosensor able to detect the immunoglobulin G as a model biomarker in blood serum (Figure 7). In their study, the authors realized an AgNPs-reduced GO composite from a heat-assisted wet chemical reduction that can be easily deposited onto a screen-printed carbon electrode and used as support for adsorption of unfunctionalized antibodies specific for immunoglobulin G (IgG). Interestingly, this very simple procedure leads to a hybrid multicomponent immunosensor with high specificity for its target being very unsensitive to 100-fold-concentrated interfering biological substances such as dopamine, glucose, interleukin-15 and myoglobin either used alone or mixed and showing a very low LOD value of 0.00086 ng mL^−1^ of IgG concentration within two working ranges, 0.001–0.05 and 0.05–50 ng mL^−1^. The authors proposed this device as a simple, low cost, high-sensitivity- and high-selectivity-method clinical diagnosis.

According to this ultrasensitive device based on IgG or its derivatives, the use of such 2D nanomaterial as a stand-alone component or combined with other nanomaterials is quickly evolving in the field of immunosensors and many other examples can be recalled, e.g.: screen-printed carbon electrodes coated with GO for detection of the epithelial sodium channels in urine [70]; GO-Fe_3_O_4_NPs-Prussian blue-AuNPs on screen-printed electrodes to detect the hepatitis B surface antibody in serum [71]; glassy carbon electrodes coated with electroreduced carboxyl G and functionalized mesoporous silica NPs-Methylene blue-AuNPs for detection of galectin-3 in serum [72]; Au colloids-TiO_2_NPs-chitosan composites embedded within carbon nanochips, the latter being a complex of 6–8 rolled sheets of G, on Au electrodes for detection of β-lactoglobulin in food [73]; 3D porous cryogels made using chitosan-G-ionic liquid-ferrocene composites and decorated with AuNPs-antibodies on screen-printed carbon electrodes to detect the prostate-specific antigen in serum [74]; and ZnONPs-polyaniline-chitosan functionalized with urease on indium tin oxide electrodes to detect urea in serum [75].

Interesting results have been published using alternative carbon allotropes such as carbon nanotubes (NTs), where the high aspect ratio guarantees large, functionalized surfaces compatible with biological probes. In this regard, Zhao et al. [76] described a method to realize a sandwich-like electrochemical immunosensor based on multi-walled carbon NTs to detect the carcinoembryonic antigen in blood serum (Figure 8). In this study, a glassy carbon electrode is coated with AuNPs via electrodeposition, which act as supporting surfaces to link a first antibody that specifically targets the carcinoembryonic antigen analytes. The second component of the sensor is built using commercial carboxylated multi-walled carbon NTs that are functionalized with CoS_2_ using the hydrothermal method and distilled aniline under reduced pressure to get a nanostructured carbon NTs-CoS_2_-polyalinine composite with free NH_2_- groups. The amine groups allow double conjugation of the composite with secondary antibodies and the horseradish peroxidase enzyme; in this configuration, the secondary antibodies recognize the carcinoembryonic antigens captured on the electrode by the first AuNPs-bound antibody probes. In the presence of hydrogen peroxide, a triple reaction occurs on the immunosensor as the hydrogen peroxide is reduced to water using electrons coming from CoS_2_ that in turn is reduced back via the horseradish peroxidase enzyme (Figure 8). Under this setup, the polyaniline bound to the composite act as a non-diffusing electroactive substance that effectively transfers the electrons from the active center of the enzyme to the surface of the underlying glassy carbon electrode; meanwhile, it also acts as a signal amplificatory molecule and increases the signal-to-noise ratio, thereby improving the stability of the device. The authors used this complex system to detect the analyte as a biomarker in clinical diagnostics showing an LOD of 0.33 pg mL^−1^ within a wide range of 0.001–40 ng mL^−1^.

In this context, other similar examples are worth mentioning for their use of both multi-walled and single-walled carbon NTs alone or complexed in nanostructures such as enzyme-immobilized single-walled carbon NTs for detection of glucose [77], vinyl ferrocene-, N-hydroxy succinimide acrylate-bifunctionalized carbon NTs conjugated with antibodies on glassy carbon electrodes to detect α-fetoprotein in serum [78] and carboxylated multi-walled carbon NTs-polyalinine composites functionalized with AuNPs-antibodies on glassy carbon electrodes for detection of prostate-specific antigen in serum [79]. To recall, other materials can be used as NTs in place of carbon allotropes such as porous hydrogen titanate NTs-glucose oxidases bioconjugates on titanium foils to detect glucose [80].

Taking advantage of 2D inorganic nanomaterials, many other examples can be recalled that extend the versatility of proteins in building advanced electrochemical biosensors. An interesting example is published by Ma et al. [81], where a sandwich-like electrochemical immunosensor is described for detecting the neuron-specific enolase as a tumor biomarker in blood serum, taking advantage of a hybrid nanostructured complex based on ultrathin MnO_2_ nanosheets (Figure 9). The authors obtained AuNPs-embedded zinc-based MOFs through heat-assisted synthesis and purified via centrifugation resulting in AuNPs coated with a 3D framework that can be applied on glassy carbon electrodes. Antibodies targeting the neuron-specific enolase analytes are activated with EDC/NHS crosslinking chemistry, allowing the COOH– groups of the biomolecules to covalently bind to the free amino groups on the AuNPs-embedded zinc-based MOFs surface. In this configuration, the embedding of AuNPs improves the conductivity of the zinc-based MOFs and accelerates the electron transfer to the underlying electrode. To further improve the electrochemical properties of the immunosensors, the authors realized colloidal trimetallic Au-Pd-Pt nanocubes under reducing conditions with the gold embedded within a shell of palladium that in turn is coated using platinum and which have been used to functionalize ultrathin MnO_2_ nanosheets. This composite can be treated via adsorption with a second antibody targeting the neuron-specific enolase and used to sandwich the modified electrodes upon capture of the analyte. The immunosensor has been used to catalyze the reduction of hydrogen peroxide and promote the oxidation of hydroquinone to quinone upon binding with the analytes molecules resulting in an extremely efficient sensing activity with an LOD of 4.17 fg mL^−1^ in a very broad working range of 10 fg mL^−1^–100 ng mL^−1^. The authors claimed the use of the sensor in ultrasensitive early clinical bioanalysis.

This example recalls similar studies about sandwich-type immunosensor with high detection efficiency including bamboo-like carbon nanostructure-toluidine blue-functionalized copper-based MOFs on glassy carbon electrodes for detecting C-reactive protein in serum [82] and antibodies-conjugated DNA dendrimers on antibodies-functionalized glassy carbon electrodes to detect the prostate-specific antigen in serum [83].

Table 2 is intended to provide a list of the current electrochemical immunosensors reported in this review and their main features.

### 2.2. Optical Biosensors

#### 2.2.1. Aptamers-Based Optical Biosensors

Like aptasensors based on coated or functionalized electrodes, devices exploiting label-free colloidal or soluble nanomaterial–biomolecule nanostructures have quickly gained attention. Indeed, working in solution enables the operator to reveal the target analyte by taking advantage of very sensitive methods such as fluorescence, surface plasmon resonance (SPR) or surface-enhanced plasmonic resonance (SERS). Colorimetric, naked-eye methods able to detect analytes with good LOD values are also employed in these cases. Note that the labelling is required to detect the signal upon target engagement but exclusively regards the nanometric probes rather than the target analyte thus keeping the advantages typical of label-free methods.

As a starting example, the study reported by Wang et al. [84] describes a simple method for detecting kanamycin in milk using quantum carbon dots (CDs) and aptamers (Figure 10). In this study, AuNPs are synthesized via heat-assisted chemical reduction and used as support for coating with kanamycin-specific unfunctionalized aptamers; the aptamers-coated AuNPs remain stable in solution as single monodispersed colloidal NPs that resist to aggregation even after the addition of increasing amounts of NaCl salt, a well-known aggregating agent for metal NPs. In this status, the AuNPs can effectively quench the intrinsic fluorescence of the quantum CDs in solution as revealed using fluorescence spectroscopy. However, when kanamycin is revealed and captured it quickly starts competing for the binding with the aptamers thus breaking the low affinity adsorption interactions with the AuNPs; in these conditions, the monodispersed aptamers–AuNPs complexes are broken and the bare AuNPs become much more sensitive to the salt leading to visible aggregation. The so-formed aggregates change their absorption spectrum and no longer quench the fluorescence of the carbon QDs resulting in the recovery of the fluorescence signal that is linearly proportional to the concertation of kanamycin revealed. This aptasensor shows an LOD of 18 nM in the working range 0.04–0.24 μM and is proposed for the detection of antibiotics in food thus being adapted for several other antibiotics relying on different aptamer probes.

Colloidal probes which tend to precipitate or aggregate upon binding to the target molecules represent a common strategy for being simple and easy to monitor with common lab equipment, such as fluorescence or UV-Vis spectrophotometers. In this regard, other similar aptasensors can be recalled, highlighting the importance and efficacy of such a strategy where the aggregation of nanomaterials triggers recordable optical changes upon detection of the analyte, for example AuNPs-mediated fluorescence of Rhodamine B to detect carbendazim in water [85] or sulfamethazine in water and soil [86], AuNPs-mediated fluorescence of aptamer-complementary SYBR Green I-functionalized cDNA to detect sulphadimetoxine in water and fish [87] and fluorescent AuNPs-carbon QDs to detect adenosine triphosphate [88].

In this context, another significative example is described by Yu et al. [89] that prepared polydopamine NSs and fluorescent aptamer probes provided with DNase-I-assisted recycling amplification to detect the matrix metalloproteinase-9 and -2 in the urine and tissue homogenate of unilateral ureteral obstruction mice (Figure 11). The polydopamine NSs are obtained via a simple chemical self-aggregation under controlled wet conditions before applying the two metalloproteinase-specific aptamers marked with two different fluorophores, i.e., 5-carboxyfluorescein and Texas Red. The adsorption of the aptamer probes is likely to occur via π-π interactions and hydrogen bonding leading to quenching of the fluorophores mediated by the neighboring polydopamine NSs based on the Förster resonance energy transfer (FRET). In the presence of the metalloproteinases, the respective aptamers become detached from the polydopamine surface leading to the release as free metalloproteinase-aptamer soluble complexes and therefore to recordable and significative fluorescence emission. Interestingly, the addition of DNase-I leads to an improvement in detection as it selectively cuts the aptamers dissociated from the polydopamine nanospheres and bound to the target analytes; this reaction in turn allows the released targets to further interact with the residual fluorescent aptamers still linked to the polydopamine surface triggering signal amplification. The authors demonstrated the efficacy of this method by detecting both the metalloproteinases, whose ratio could be considered as a potential indicator for evaluating the onset of renal interstitial fibrosis. The measured LOD values are 9.6 pg mL^−1^ (metalloproteinase-9) and 25.6 pg mL^−1^ (metalloproteinase-2) within the detection ranges of 24–600 pg mL^−1^ (metalloproteinase-9) and 64–1600 pg mL^−1^ (metalloproteinase-2). This strategy can be adapted to other nanomaterials such as the aptamer-coated AuNRs for enzyme-assisted detection of adenosine triphosphate, thus justifying its versatility and usefulness [90].

Besides fluorescence-based detection to capture analytes through aptamers-functionalized nanomaterials, other easier colorimetric methods have been developed alongside allowing the biosensing without the need of instrumentations. Indeed, colorimetry detection comes with changes in the optical properties of nanomaterials when they move from non-aggregated, monodispersed nanometric objects to clusters that aggregate and/or precipitate in solution, which usually show a red shift of their absorbance spectrum in the visible region [91]. As a typical study, Soongsong et al. [92] reported on the realization of a facile colorimetric aptasensor to detect chlorpyrifos based on polyethyleneimine-induced aggregation of AuNPs (Figure 12). In their work, the authors obtained citrate-capped AuNPs using heat-assisted chemical synthesis in wet conditions and tested their aggregation upon the addition of various amounts of polyethyleneimine that is known to displace the citrate ions coating the AuNPs surfaces leading to their aggregation because of the loss of electrical repulsion in solution. The addition of chlorpyrifos-specific aptamers triggers the formation of aptamers–polyethyleneimine complexes through electrostatic bonds thus sequestering the organic polymer and avoiding the aggregation of the AuNPs that look like red colored colloids. When chlorpyrifos is added, its affinity towards the aptamer probes breaks the complexes formed by the aptamer and polyethyleneimine molecules, the latter becoming free to adsorb over the gold surface, leading to aggregation and an evident color change from red to blue through intermediate colors depending on the amount of analyte detected. It is important to recall that this simple aptasensor is very selective for chlorpyrifos, being almost blind to many other interfering compounds including divalent metal ions and biological molecules such as glucose, urea and oxalate. The authors claimed this device as capable of determining the presence of chlorpyrifos in several foods such as water, pomelo, and longan with an LOD of 7.4 ng mL^−1^ within a range 20–300 ng mL^−1^.

Such an easy, simple method based on naked-eye colorimetric detection opens up many other possibilities as reflected by several similar examples published, e.g., AuNPs coated with unfunctionalized polyadenine aptamers to detect prostate specific antigen in serum [93]; AuNPs coated with unfunctionalized truncated forms of aptamers to detect bisphenol A in food and water [94]; AuNPs coated with unfunctionalized aptamers to detect chloramphenicol and tetracycline in food [95]; AuNPs coated with unfunctionalized aptamers for detection of *Bacillus carboniphilus* on bacterial biofilms [96]; AuNPs coated with unfunctionalized RNA aptamers for detection of human papillomavirus type 16 L1 in clinical and vaccine samples [97]; and AuNPs coated with unfunctionalized aptamers to detect acetamiprid in food and water [98].

Other colorimetric approaches based on the aggregation of NPs have been developed to realize a ready-to-use and portable device with quick naked-eye recordable results. An example worth mentioning is the aptasensor proposed by Abedalwafa et al. [99] as the supporting material is a portable strip rather than an electrode or colloidal solution. The device has been obtained by combining an electrospinning fabrication of nanofibrous (NF) membranes and a wet colloidal synthesis of NPs for colorimetric detection of the kanamycin (Figure 13). The authors grafted glutamic acid molecules into strips made by electrospun NF membranes of cellulose acetate to obtain COOH-functionalized NFs; these have been chemically modified through a classical EDC/NHS crosslinking coupling reaction that allows NH_2_-aptamers targeting kanamycin to be linked to the NF membranes using amide bond. As a probe, AuNPs are synthesized chemically and coated via adsorption with a complementary single-stranded DNA (cDNA) of the kanamycin aptamer and left to hybridize with the functionalized NF membranes by taking advantage of the hydrogen-based hybridization between the two nucleic acid strands. Because the aptamer shows higher affinity to the kanamycin molecules than to its cDNA, in the presence of antibiotic the cDNA is replaced, leading to detachment of the cDNA-decorated AuNPs probes from the strips; bleaching is observed using a visual color inspection and UV-Vis spectrophotometry. The authors used this aptasensor to detect kanamycin residues in food samples with a tested LOD of 2.5 nM in the working range of 2.5–80 nM. Based on the same principle, similar devices can be found as those reporting on AuNPs functionalized with truncated aptamers run onto a nitrocellulose membrane through lateral flow assay to detect oxytetracycline in food [100] and AuNPs coated with unfunctionalized aptamers run onto nitrocellulose membrane on a paper-based microfluidic device to detect gentamicin in food [101].

The ease of making these kinds of colloidal optical sensors comes along lower LOD values, therefore showing decreased sensitivity when compared with the electrochemical counterparts. However, like the electrochemical aptamers based on voltametric measurements, the combination of colloidal probes with more efficient nanomaterials helps improve the performance of the resulting aptasensor as reported in many cases. The first example to be recalled is described by Chinnappan et al. [102] and relies on a basic improvement where GO is used as a platform for fluorescent aptamers and detection of tropomyosin, a major shrimp allergen, using fluorescence spectroscopy (Figure 14). The authors investigated the method by using several fluorescein-tagged aptamers differing for the length and provided a simple method based on GO-driven quenching to detect the analyte in solution. Namely, it is known that 2D nanomaterials interacts with both double- and single-stranded DNA sequences via non-covalent π-π stacking and hydrophobic interactions [103]; in this condition, when fluorescent aptamers interact with GO, the fluorophore is readily quenched so long as the nucleic acid is attached. In this example, a truncated fluorescent fluorescein–aptamer conjugate is used as a recognizing probe and its fluorescence is shown to increase linearly via the addition of an increasing concentration of tropomyosin, likely because the target molecules upon interaction with the truncated aptamer trigger the fluorescence signal by detaching from the GO surface to the solution state. Interestingly, the authors claimed that the truncated form of the fluorescent aptamer results are more sensitive than longer sequences, presumably due to (i) the elimination of non-essential nucleotides which may not be involved in capturing the target and (ii) their involvement in the formation of the aptamer-tertiary structure that reduces the chance of binding to tropomyosin. The authors showed that the aptasensor is provided with an LOD of 2 nM within a range of 0 to 1.3 μM to detect the tropomyosin allergen from shellfishes as well as chickens.

Accordingly, several similar studies have been described showing the advantage of having a nanometric 2D platform for fluorescent or colorimetric detection of analytes, e.g., enzyme-assisted, MnO_2_ nanosheets-based AuNPs aptasensor to detect the activity of the alkaline phosphate and the mycotoxin ochratoxin A in food [104], Fe_3_O_4_-GO-assisted AT-rich three-way junctions DNA-stabilized CuNPs to detect isocarbophos in food and water [105] and zirconium-porphyrin MOFs coated with fluorescein-conjugated aptamers to detect chloramphenicol in food [106].

Table 3 provides a list of the current optical aptasensors reported in this review and their main features.

#### 2.2.2. Proteins-Based Optical Biosensors

As reported for aptamers, proteins, antibodies and enzymes are very attractive when combined to nanomaterials and build colloidal or soluble sensors with special optical features. If compared to single-stranded DNA or RNA probes, the protein surface available for functionalization is much more feasible and enables multiple chemical functionalities to coexist on the same molecule simultaneously. The surface versatility makes proteins ideal building blocks for realizing complex nanostructures soluble in hydrophilic media even when combined to hydrophobic nanomaterials.

A proper example is described by García-Rubio et al. [107] where AuNPs are conjugated to antibodies to detect the epithelial sodium-channel protein present on platelets through simple, visible spectroscopy (Figure 15). The immunosensor is obtained using commercial AuNPs reacted with 3-mercaptopropionic acid in water to ensure complete functionalization of the metal surface through thiol-gold bonds thus leaving many -COOH groups accessible for further functionalization. In this condition, antibodies specific for the epithelial sodium-channel protein can be covalently linked to the functionalized gold surface by taking advantage of the EDC/NHS crosslink method resulting in stable hydrophilic AuNPs–antibodies conjugates with optical absorption in the visible spectrum. Upon binding with the target analyte, i.e., the epithelial sodium channel, a secondary fluorescent antibody can be used to detect the complex formed. In these conditions, the samples containing the epithelial sodium channel treated with AuNPs-antibodies probes show a much larger intensity if compared to samples treated only with anti-epithelial sodium channel antibodies; this result is likely due to the plasmonic effect occurring on the surface of the metal NPs that affects the optical properties of the neighboring fluorophores [108]. To note, in this configuration the distance between the antibodies and the metal surface provided by indirect conjugation through 3-mercaptopropionic acid as a short ligand is propaedeutic for avoiding the quenching effect that is induced by the plasmonic nanostructures at very small distances from the fluorophore. The authors used their immunosensor to discriminate between hypertensive and normotensive individuals, which represents a useful diagnostic tool for arterial hypertension.

According to the same simple strategy, many other nanometric immunosensors have been proposed that exploit NPs conjugated with antibodies, as for instance magnetic Au-coated Fe_3_O_4_NPs-antibodies for detection of glyphosate in tap water [109]; AuNPs-antibodies and Fe_3_O_4_NPs-antibodies for detection of prostate-specific antigen in serum [110]; AuNPs conjugated with three fluorophore-modified proteins BSA, rhodamine B and β-lactoglobulin to detect nephrin and podocin in urine [111]; and magnetic AuNPs-antibodies coupled to SPR-active sensor disks for detection of the CD5 biomarker in serum [112].

The enhancing or quenching effect of nanomaterials based on their plasmonic behavior is found to be useful, taking advantage of 2D nanostructures as reported for GO sheets. GO, similar to metal NPs, triggers significant changes on the physical properties of molecules thus becoming a suitable tool for detecting low concentrations of analytes via optical methods. In their study, Ortiz-Riaño et al. [113] demonstrated the possibility of detecting the prostate-specific antigen in urine by taking advantage of GO-induced quenching of fluorescent QDs-functionalized antibodies (Figure 16). The authors used multi-well polystyrene amino-functionalized black plates as measuring platforms where each well is filled with cationic poly-lysine glass slides which act as support for the immobilization of GO. This is obtained by adding GO solutions within the wells and allowing the sheets to adsorb onto the glass surface via electrostatic interactions relying on the positively charged poly-lysine molecules and the negatively charged GO surface. As a probe, biotin-conjugated antibodies targeting the prostate-specific antigen are coupled to fluorescent streptavidin-conjugated QDs by taking advantage of the strong biotin–streptavidin interaction. The resulting QDs-antibodies complex allows the target analyte to be captured in solution without perturbing the fluorescence of the QDs. This immunosensor is realized in such a way to quench the fluorescence of the QDs-antibodies probes quickly and strongly in the absence of prostate-specific antigen molecules upon their adsorption onto the surface of the underlying GO substrate; however, when the probes capture the analytes, the resulting complex is weakly detached due to the low affinity between the GO-coated surface and the relatively long distance between the QDs and the GO, thus resulting in increased signal detection. To note, the authors used their immunosensor as a useful device for several clinically relevant analytes having an LOD value of 0.05 ng mL^−1^ with a working range of 0.15–10 ng mL^−1^, thus proving to be much more sensitive than commercially available immunoassays even without the use of fluorescence-enhancing nanomaterials.

The quenching effect induced via 0D and 2D nanomaterials is a very common tool exploited in immune-based sensing and is reported by other studies, as for instance antibodies-conjugated CdSe-ZnSQDs and AuNRs to detect porcine reproductive and respiratory syndrome virus in swine serum [114] and rhodamine-labelled affinity peptides adsorbed onto WS_2_-Pt-Fe_2_O_3_ polycaprolactone micromotors for detection of bacterial lipopolysaccharides [115].

Optical immunosensors based on papers and pads for visive colorimetric and/or fluorescence detection are also reported. An example describes how hydrophobic QDs can be coupled to protein moieties to increase both the solubility and fluorescence and detect glycosylated hemoglobin in blood samples as reported by Li et al. [116] (Figure 17). In this study, wet chemical synthesis of oleic acid-capped CdSe-ZnSQDs is realized and the resulting material is mixed in chloroform with bovine serum albumin under ultrasonication conditions at various QDs:protein ratios. The interaction between QDs and the protein moieties is likely to occur through the cysteines and the disulfide bonds contained in the amino acid sequence, which preferentially replace the hydrophobic ligands on the surface of the CdSe-ZnSQDs. Upon evaporation to remove the organic solvent, the bovine serum albumin-wrapped QDs nanobeads readily dissolve in water, accounting for the hydrophilic behavior of the abundant water-soluble groups at the protein surface that also allow a thicker silica (SiO_2_) shell to grow. The silica shell is obtained by mixing the nanobeads in ammonia and tetraethyl orthosilicate before the addition of (3-aminopropyl)triethoxysilane to produce the silica surface functionalized with -NH_2_ groups. Following the treatment with glutaric anhydride, the silica-coated nanobeads are decorated with many -COOH functional groups that allow sheep anti-human hemoglobin polyclonal antibodies to be covalently linked onto the surface through EDC/NHS crosslinking chemistry, resulting in the final fluorescent bioconjugated QDs sensors. The same procedure is applied to create fluorescent QDs conjugated to rabbit anti-deoxyribo-nucleoprotein antibodies. The authors prepared a complex device for lateral-flow immunoassay using a nitrocellulose membrane as a physical support, which is divided into a sample loading zone and a revelation zone. Both the bioconjugated QDs are sprayed onto the absorbent nitrocellulose membrane; similarly mouse anti-human hemoglobin and anti-deoxyribo-nucleoprotein-bovine serum album antibodies are adsorbed onto the revelation zone. In the presence of hemoglobin, a complex is formed with the QDs conjugated with anti-human hemoglobin polyclonal antibodies that leave the adsorption zone and migrate towards the revelation zone forming a sandwich-like structure with the mouse anti-human hemoglobin molecules and resulting in a clear luminescent band onto the nitrocellulose membrane; conversely, without hemoglobin molecules the sandwich structure is not formed and only the control luminescent band is revealed due to the formation of a complex between the QDs conjugated with the rabbit anti-deoxyribo-nucleoprotein antibodies and the anti-deoxyribo-nucleoprotein-bovine serum album antibodies adsorbed onto the revelation zone. The device has been proposed as a signal-amplification probe for a lateral-flow immunoassay biosensor for the accurate detection of biomarkers such as the glycated hemoglobin that is considered as the gold standard for long-term glycemic control in diabetes mellitus.

Further examples must be recalled that take advantage of using proteins and nanomaterials coupled to optical methods and or characteristics to reveal analytes, as for instance those reported for a multiplex lateral flow lateral flow assay device onto nitrocellulose membrane and antibodies-functionalized AuNSs for detection of metalloproteinase-9, S100 calcium-binding protein B and neuro-specific enolase in blood [117], antibodies-functionalized magnetic Fe_3_O_4_ NPs capped with a gold shell to detect the human growth hormone in plasma [118] and antibodies-conjugated Mn-ZnS QDs for analyte-induced gold amplification for detection of prostate-specific antigen in serum [119].

Table 4 provides a list of the current optical protein-based sensors reported in this review and their main features.

## 3. Detection of the SARS-CoV-2 Virus: A Practical and Recent Paradigm

Prevention of diseases in terms of tracing and monitoring represents the evergreen desire to take care of all pathological conditions, from cancers to infectious diseases, especially in cases of highly contagious organisms.

In this context, the recent and still current diffusion of the SARS-CoV-2 virus and its derivatives that quickly spread worldwide from January 2020 represents the prime example for at least two main reasons: first, infectious diseases usually move across the early to late stages of development inside the host organism by initially being latent and difficult to detect before becoming aggressive and causing severe health conditions; secondly, the development of pharmaceutical therapeutics takes time, usually years, to achieve final approval by regulatory institutes. Throughout 2022, local, national and global initiatives such as the World Health Organization (WHO) have unceasingly expressed main concerns regarding the SARS-CoV-2 pandemics especially related to its variants. The WHO, for instance, placed the COVID-19 disease within the list for “Prioritizing diseases for research and development in emergency contexts”, which also include pathologies caused by Ebola and Zika amongst many others [120] while updating the tracking list of all SARS-CoV-2 variants [121]. Detection of this kind of microorganism is currently based on molecular or serological assays, i.e., the polymerase chain reaction (PCR), which take time to get the result or immunological assays aimed at revealing the antibodies produced by the host organism during both the early and later stages of the infection. Finding new materials to improve the LOD and sensitivity is therefore the future in terms of advances in prevention and healthcare. As a matter of fact, the literature has been strongly prompted with hundreds of immunosensors for SARS-CoV-2 antigens rather than aptasensors.

In such a huge and broad context, it now becomes clear that methods for preventing highly diffusive diseases through detection and monitoring of the etiological agents, even at the very early stages, are the missing weapons we need to realize. The following examples are not intended to comprehensively cover the relentless growth of papers reporting on detection strategies for SARS-CoV-2 but instead aim at highlighting how very different detection strategies can be used for sensitive label-free detection based on bioconjugated nanomaterials. It must be noted that the examples described herein refer to devices able to detect whole, intact viral particles rather than single, isolated molecular components of the organism. A proper starting study is described by Shao et al. [122], a study that reported on an electrochemical immunosensor that includes a field-effect transistor (FET) based on high-purity semiconducting single-walled carbon NTs to assess the presence of SARS-CoV-2 particles in clinical nasopharyngeal samples (Figure 18). The authors prepared interdigitated Au electrodes patterned on Si/SiO_2_ by means of photolithography and containing micrometric channels. These electrodes are modified through dielectrophoresis with the semiconducting single-walled carbon NTs and annealed at a high temperature before the use. The carbon NTs act as supporting nanomaterials to covalently attach antibodies targeting the SARS-CoV-2 spike protein or the nucleocapsid protein to the FET via the EDC/NHS coupling chemistry. The authors tested the efficacy of both the spike- and nucleoprotein-specific immunosensors on remnant nasopharyngeal swab samples by adding a few microliters to the antibody-functionalized devices and incubating a few minutes before rinsing with water to remove unbound virus particles. To note, the electrochemical FET performances in water show that these immunosensors can detect SARS-CoV-2 viruses very quickly, within 5 min, using both the anti-spike and anti-nucleocapsid protein antibodies-functionalized carbon NTs in clinical samples without prior sample processing. To note, the FET device functionalized with the anti-spike antibodies performed better than those functionalized with anti-nucleocapsid antibodies in discriminating positive and negative nasopharyngeal samples and resulted in being less susceptible to nonspecific species. The authors reported very low LOD values of 0.55 fg mL^−1^ and 0.016 fg mL^−1^ for the anti-spike and the anti-nucleocapsid antibodies devices, respectively, therefore being suitable for clinical diagnostic applications in detecting virus particles or for realizing multiplex devices for other kinds of infections.

It must be recalled that other carbon allotropes can be exploited as supports for antibodies bioconjugation and the realization of advanced immunosensor devices. Similar FET devices, for instance, can be obtained by taking advantage of 2D graphene sheets conjugated with antibodies and used to detect viral particles without sample pre-treatment or labelling [123].

When speaking about ultrasensitive methods for detection of analytes, Raman spectroscopy stands as one of the most useful and straightforward technique. In this context, the SERS effect that occurs on nanostructured surfaces is highly desired, for it can amplify the signal coming from the analyte or the probes in contact with the nanostructure. A study relying on this approach is reported by Zhang et al. [124] that presented the use of self-assembling layers of uniform AuNPs showing plasmonic-based SERS activity to detect SARS-CoV-2 particles contained in saliva (Figure 19). In this study, SERS tags are prepared starting from silver NPs (AgNPs) synthesized in wet conditions via heat-assisted chemical reduction and functionalized on the surface with mercaptobenzoic acid. The resulting mercaptobenzoic acid-capped AgNPs are used as supports to immobilize anti-SARS-CoV-2 spike antibodies serving as probes for the viral particles. The authors realized the SERS-immune substrate using AuNPs synthesized through a seed-mediated growth method from a mixture of cetyltrimethylammonium chloride, ascorbic acid and Au seeds followed by treatment with the oil/water/oil, three-phase liquid–liquid interfaces self-assembly method that allows the AuNPs to self-assemble into a monolayer film with a metallic screen. This structure can be deposited over hydrophilic-treated silicon wafers, which can then be treated with mercaptoundecanoic acid serving as support in immobilizing the anti-SARS-CoV-2 spike antibodies. This configuration is adopted to mimic the ELISA method of detection of analytes according to a sandwich-like device. Briefly, a few microliters are applied onto the SERS-immune substrate of functionalized AuNPs before being sandwiched with the SERS nanotags of functionalized AgNPs. This device has been shown to detect the SARS-CoV-2 spike protein in both saliva and buffer with LOD of 6.07 fg mL^−1^ and 0.77 fg mL^−1^, respectively, as well as in serum and blood with LOD of 7.60 fg mL^−1^ and 0.10 pg mL^−1^, respectively. The ability of this immunosensor to detect whole viral particles from saliva samples extruded from COVID-19 patients might hold promising potential for the sensitive screening of symptomatic and asymptomatic individuals during early infection.

Accounting for the plasmonic behavior typical of metal nanostructured objects, other interesting approaches have been developed for the ultrasensitive detection of SARS-CoV-2 particles from real samples. Coupling between plasmons and optical fibers is one of the most intriguing as the light travelling across an optical fiber is not affected by electromagnetic interference and, as reported for U-shaped configurations, they are the most efficient to achieve localized plasmon response and provide the quickest response. In this context, the immunosensor described by Hadi et al. [125] and based on U-shaped optical fibers is worth mentioning; it is a simple but very effective device for the ultrasensitive detection of viral particles from real samples such as oropharyngeal or nasopharyngeal swabs (Figure 20). The authors realized a plastic optical fiber consisting of a polymethylmethacrylate core and fluorinated polymer cladding bent at the middle portion to get a U-shaped configuration fixed via heat treatment. The resulting structure is selected to have a high refractive index at the U-shaped portion of about 3.1. The resulting U-bent fibers are treated using acid hydrolysis and obtain –COOH groups on the polymethylmethacrylate core followed by treatment with hexamethylenediamine that creates -NH_2_ functional groups on the surface. Further treatment with silane and glutaraldehyde is required for a covalent immobilization of the while fiber structure. As plasmonic elements, AuNPs are chemically synthesized by reducing agents before being immobilized on the U-bent fiber optic probe and used as supports to covalently conjugate antibodies specific for the SARS-CoV-2 nucleoprotein. The antibodies-immobilized probes are then treated with bovine serum albumin to avoid nonspecific interactions between the viral particles and the AuNPs. In this configuration, when the AuNPs-bound antibody probes are exposed to the virus they undergo a modulation of the light transmitted and passing through the U-shaped fiber; during this process, the underlying AuNPs can amplify the interaction signal that enhanced light interaction resulting in an increase in intensity. The authors used their device to be very cost effective with minimum logistics and without the need of trained personnel. Furthermore, they argued that the device can be simply connected to a national health services system for reporting, self-analysis and limiting the COVID-19 outbreak.

## 4. Discussion and Conclusions

Label-free detection based on hybrid nanostructured materials where biological moieties are used as probes is rising as one of the main strategies to achieve this goal. This review summarized several examples of label-free devices containing biomolecular components, mainly nucleic acids and proteins, and nanostructured materials with 0D to 2D shape, highlighting how both the electrochemical and optical measurements can reach ultrasensitive detection, from molecules to whole organisms, even at very low LOD values down to femto moles (10^−15^ moles) of analyte to be detected. Many of the reported examples show very high selectivity for the target even when tested in real complex media such as serum, natural water, urine and saliva, and do not require pre-treatment of the sample therefore helping the realization of simple, ready-to-use devices; furthermore, many of these examples can realize detection within a few minutes.

Aptamers and immunoglobulins represent the two main biological components integrated in such devices for they highly ensure selectivity with no or little off-target engagement. It seems, however, that when approaching attempts of detection of viral particles, such as SARS-CoV-2, the immunosensors, i.e., devices using antibody probes, are the most exploited; conversely, aptasensors are still in a childhood stage and further development is needed even though examples are already demonstrating their efficacy for accurate and sensitive detection [126]. However, accounting for the high rate of mutation that the SARS-CoV-2 spike protein undergoes [127] and the intrinsic structural disorder shown by the nucleoprotein [128], it is shown that the development of multiplexed sensors with integrated detection systems might be needed to overcome the risk of not detecting mutant viral particles. As a matter of fact, in general the number of label-free immunosensors and aptasensors is quickly growing in terms of published scientific papers and reviews as collected, for instance, in the Scopus database (Figure 21), and the global market size of biosensors valued at 24.9 billion USD in 2021 is expected to expand at a growth rate of 8.0% from 2022 to 2030 [129], therefore proving the importance of such a spreading and flourishing area of applied science.

Now as never before, the importance of detection, tracing and monitoring activities are particularly relevant and are targeted worldwide, and the interest in obtaining advanced devices to achieve the quick and highly performing sensing of molecules and organisms grows year by year. The environmental changes, pollution and the rising of new lethal diseases caused by etiologic agents represent primary challenges for many countries. As an example, the European Commission recently adopted a set of proposals known as “Green Deal” whereby healthy and affordable foods as well as fresh air, clean water, healthy soil and biodiversity represent some of the beneficial effects it wants to achieve by taking advantage of the “NextGenerationEU Recovery Plan” and the seven-year budget Research & Innovation programme 2021–2027 “Horizon Europe” [130]. To get an idea, the latter is supporting the research community in countries belonging to the European Union with 53.5 billion €, where one-third is reserved for the “Health” and “Food, Bioeconomy, Natural Resources, Agriculture & Environment” clusters [131]. Other countries are moving the same way, and the concept of monitoring and sensing foods, environments and biological agents is undoubtedly supposed to grow and become a cultural aspect that we cannot overlook for the next few years. This cultural scene pertaining to the development of new, advanced sensors and biosensors is therefore likely to become a prominent area for the applied sciences.

## Figures and Tables

**Figure 1 sensors-23-00949-f001:**
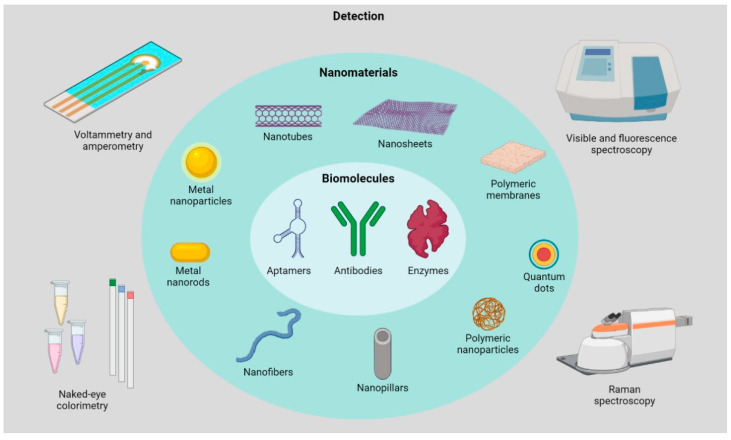
Schematic overview of the main biological, organic and inorganic components and detection strategies reported in this review. The image has been created with BioRender.com.

**Figure 2 sensors-23-00949-f002:**
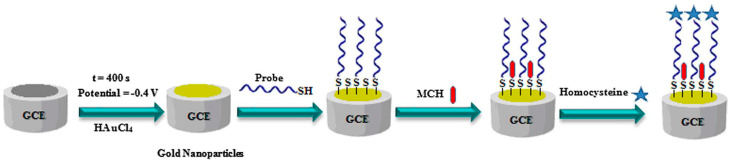
Electrochemical aptasensor for detection of homocysteine in blood serum and urine. A glassy carbon electrode is coated with AuNPs, which in turn are functionalized with SH-aptamers. The aptamers selectively bind to homocysteine leading to increased voltametric and current values. Image adapted from [25] with permission of Elsevier, 2020.

**Figure 3 sensors-23-00949-f003:**
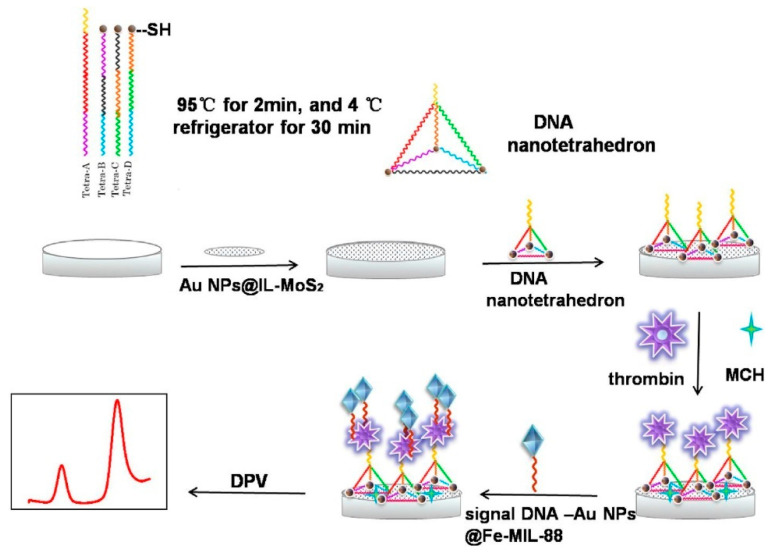
Electrochemical aptasensor for thrombin detection in blood serum. A glassy carbon electrode is coated with a AuNPs-IL-MoS_2_ composite and functionalized with SH-DNA nonotetrahedrons that bind thrombin molecules and serve as supports for successive binding of AuNPs-Fe-MIL-88 MOFs. Image adapted from [28] with permission of Elsevier, 2020.

**Figure 4 sensors-23-00949-f004:**
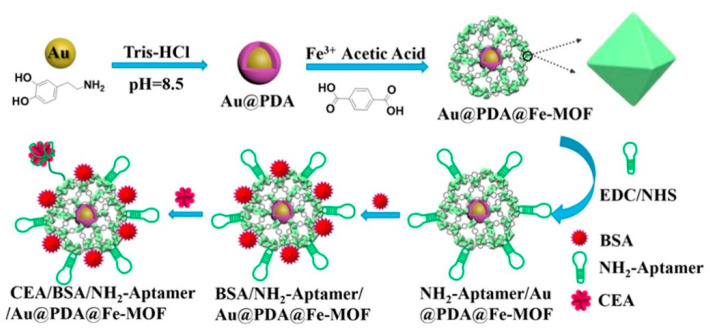
Electrochemical aptasensor for carcinoembryonic antigen in serum. A glassy carbon electrode is coated with AuNPs where Fe-MOFs and polydopamine are formed. NH_2_-aptamers for the carcinoembryonic antigen can be attached and the resulting coated AuNPs are used to detect carcinoembryonic antigen molecules. Image adapted from [39] with permission of American Chemical Society, 2020.

**Figure 5 sensors-23-00949-f005:**
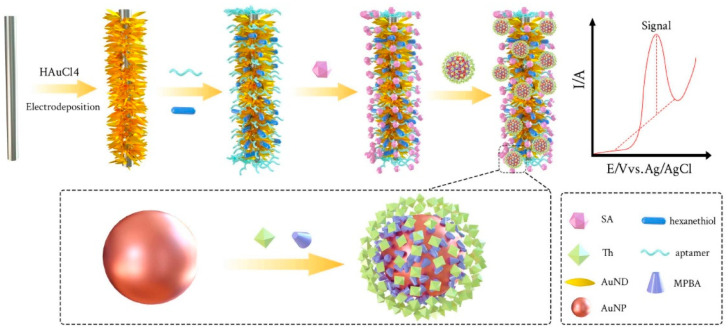
Electrochemical aptasensor for sialic acid in serum. Carbon electrode coated with Au nanodendrites and aptamers are used to sense sialic acid molecules whose presence is amplified by taking advantage of 4-mercaptophenylboric acid-thionine-functionalized AuNPs. Image adapted from [54] with permission of John Wiley and Sons, 2020.

**Figure 6 sensors-23-00949-f006:**
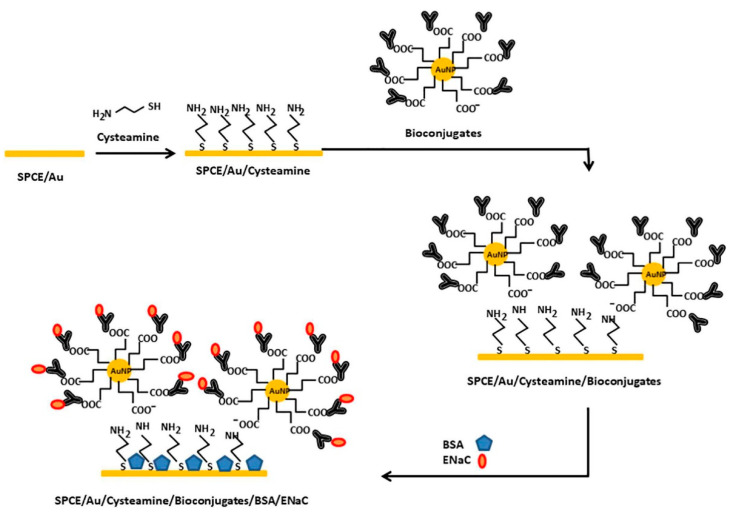
Electrochemical immunosensor for detection of the epithelial sodium channel protein in urine. Au-screen-printed carbon electrodes are used to immobilize AuNPs–antibodies bioconjugates targeting the epithelial sodium channel and used in voltametric measurements. Image adapted from [62] with permission of Elsevier, 2020.

**Figure 7 sensors-23-00949-f007:**
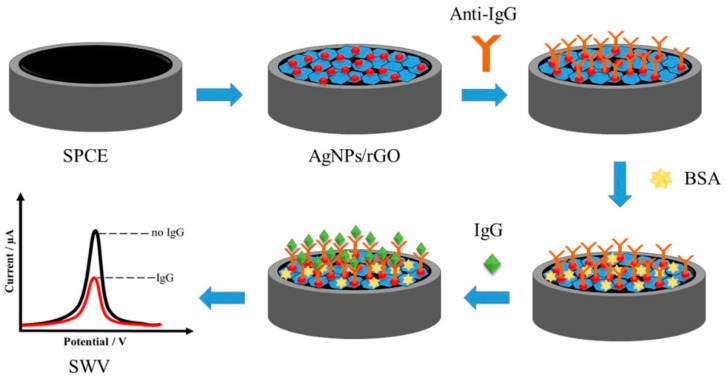
Electrochemical immunosensor for immunoglobulin G in blood serum. Screen-printed carbon electrodes are coated with AgNPs-reduced GO composites linked to antibodies specific for immunoglobulin G. Image adapted from [69] with permission of Frontiers, 2021.

**Figure 8 sensors-23-00949-f008:**
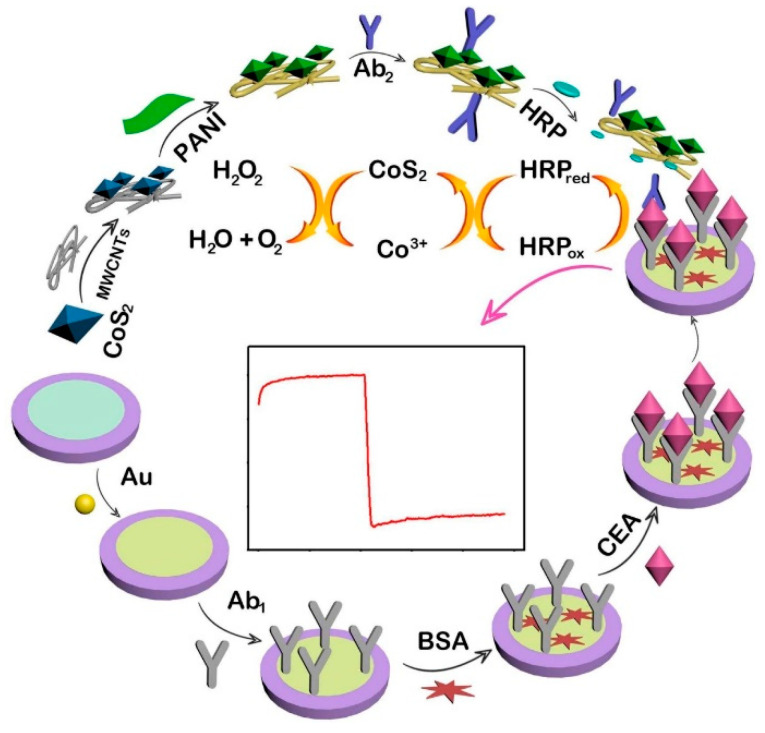
Electrochemical enzyme-assisted immuneosensor for carcinoembryonic antigen in blood serum. Glassy carbon electrodes are coated with AuNPs–antibodies conjugates which capture the target; the target in turn is sandwiched between a complex carbon NTs-CoS_2_-polyaniline-antibodies-horseradish peroxidase material that increases the electrochemical detection. Image adapted from [76] with permission of Royal Society of Chemistry, 2020.

**Figure 9 sensors-23-00949-f009:**
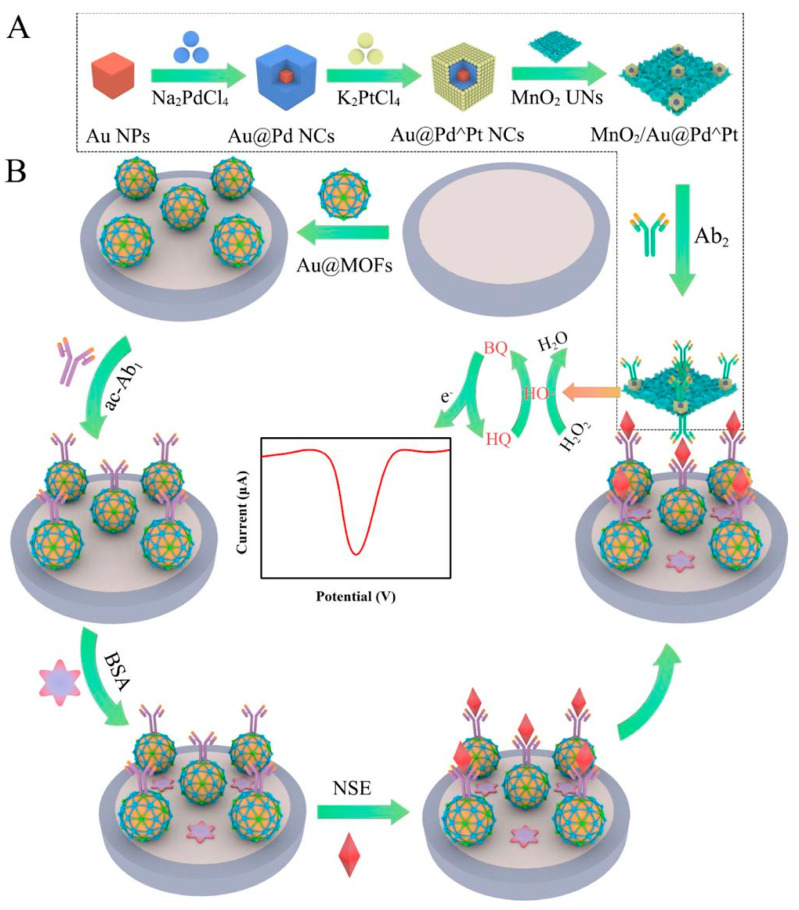
Electrochemical immuneosensor for neuron-specific enolase in blood serum. (**A**) A hybrid structure is formed using MnO_2_ nanosheets decorated with trimetallic Au-Pt-Pd nanocubes and conjugated to a secondary antibody. (**B**) Glassy carbon electrodes are coated with antibodies-functionalized AuNPs-embedded MOFs and used as support for the targeting antibodies probes. This sandwich-like architecture increases the electrochemical efficacy upon binding to the target molecules. Image adapted from [81] with permission of American Chemical Society, 2020.

**Figure 10 sensors-23-00949-f010:**
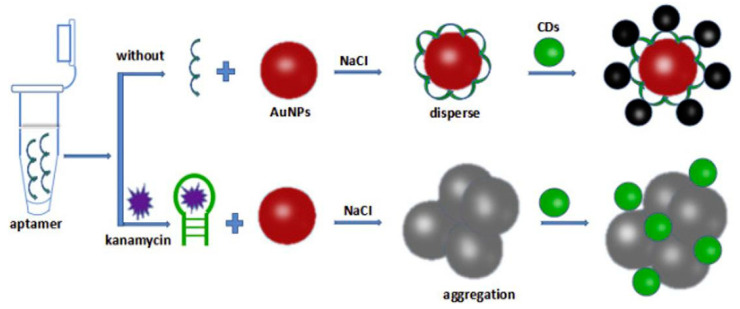
Optical aptasensor for kanamycin detection in milk. Monodispersed AuNPs coated with kanamcin-specific aptamers undergo aggregation upon displacing of the aptamers by the kanamycin molecules leading to changes in the fluorescence yield of quantum CDs. Image adapted from [84] with permission of Elsevier, 2020.

**Figure 11 sensors-23-00949-f011:**
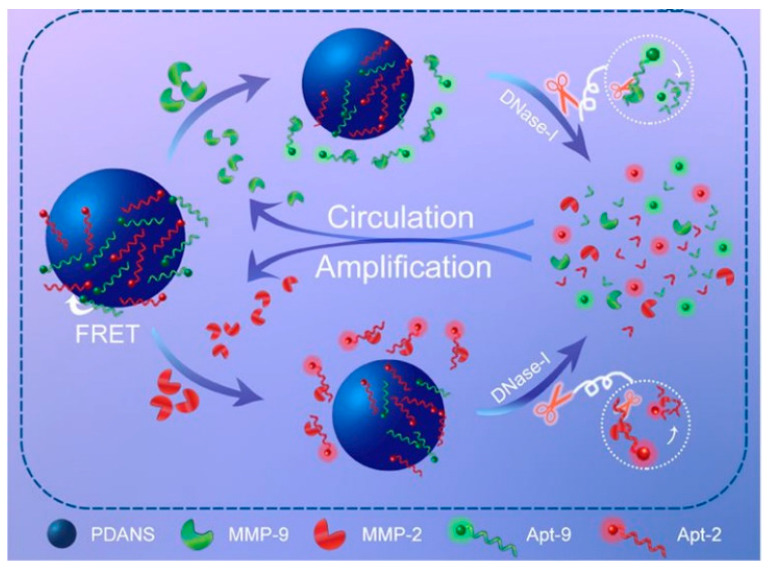
Optical aptasensors for matrix metalloproteinase-9 and -2 in urine and tissue homogenate. Colloidal polydopamine NSs coated with fluorescent aptamers are reacted with the metalloproteinases whose fluorescence is quenched. Following the binding with the targets, the aptamers detach and recover the fluorescence; in these conditions, DNase-I cuts the detached aptamers and leads to fluorescent amplification. Image adapted from [89] with permission of American Chemical Society, 2020.

**Figure 12 sensors-23-00949-f012:**
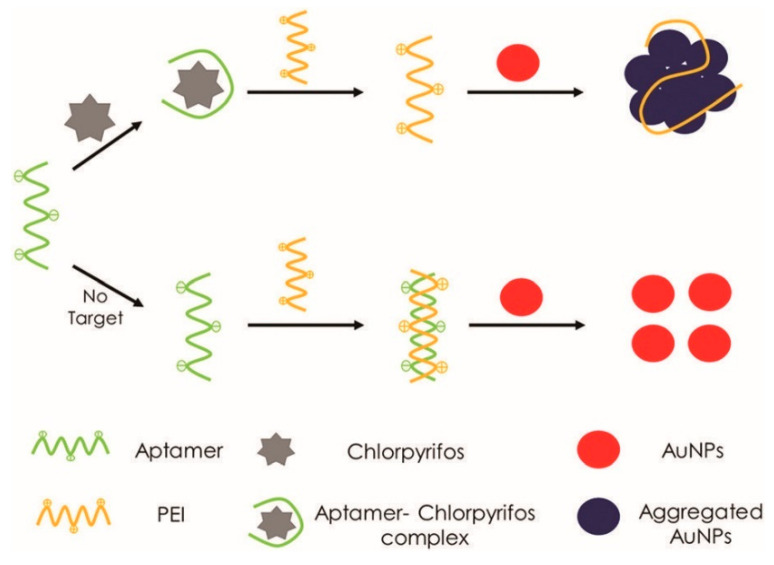
Colorimetric aptasensor for chlorpyrifos detection in water, pomelo and longan. Chlorpyrifos-specific aptamers prevent the aggregation of colloidal AuNPs by sequestering the aggregating agent polyethyleneimine. Upon detection of chlorpyrifos, the aptamer-polyethyleneimine are broken and the free polyethyleneimine molecules adsorb onto the metal surface leading to aggregation and a color change from red to blue. Image adapted from [92] with permission of Royal Society of Chemistry, 2021.

**Figure 13 sensors-23-00949-f013:**
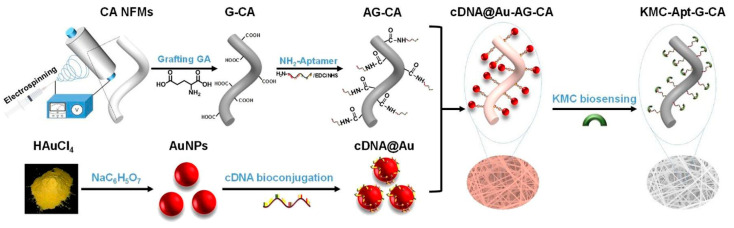
Colorimetric aptasensor for kanamycin detection in water and milk. Portable glutamic acid-grafted cellulose acetate NF membranes functionalized with kanamycin aptamer and hybridized with cDNA-coated AuNPs, the probes, react to detect the analyte, i.e., kanamycin, leading to disassembly of the aptamer–cDNA complex and release of the probes, which can be visualized as bleaching of the strips. Image adapted from [99] with permission of Springer Nature, 2020.

**Figure 14 sensors-23-00949-f014:**
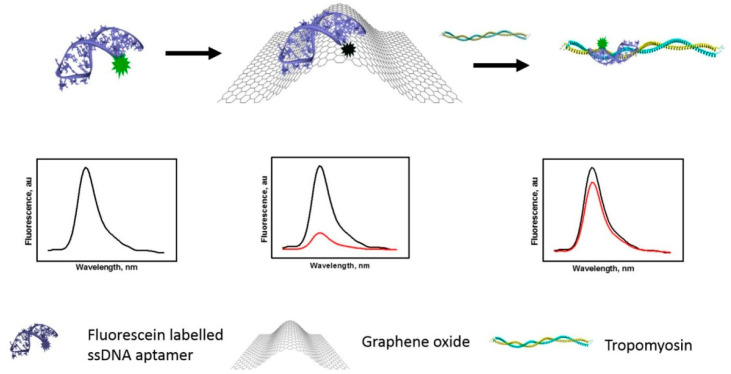
Optical aptasensor for detection of tropomyosin in food. Fluorescein-tagged aptamers are adsorbed on GO and used to detect tropomyosin according to a GO-induced quenching. When tropomyosin binds to the aptamers, they detach from the carbon surface to the solution and recover the fluorescence signal. Image adapted from [102] with permission of Elsevier, 2020.

**Figure 15 sensors-23-00949-f015:**
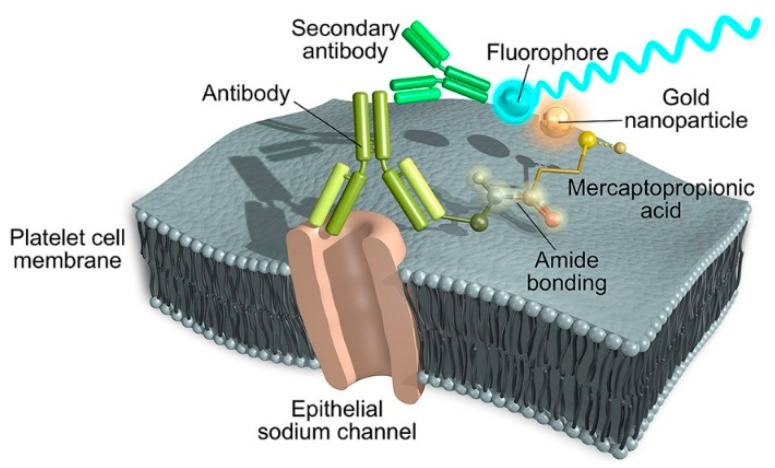
Optical immunosensor for epithelial sodium channel in blood. Fluorescein-tagged secondary antibodies are directed to epithelial sodium channel which is captured using colloidal AuNPs-antibodies conjugates. When the analyte is detected, the fluorescent antibodies are revealed and the signal is enhanced by taking advantage of the AuNPs-induced plasmonic effect. Image adapted from [107] with permission of Elsevier, 2020.

**Figure 16 sensors-23-00949-f016:**
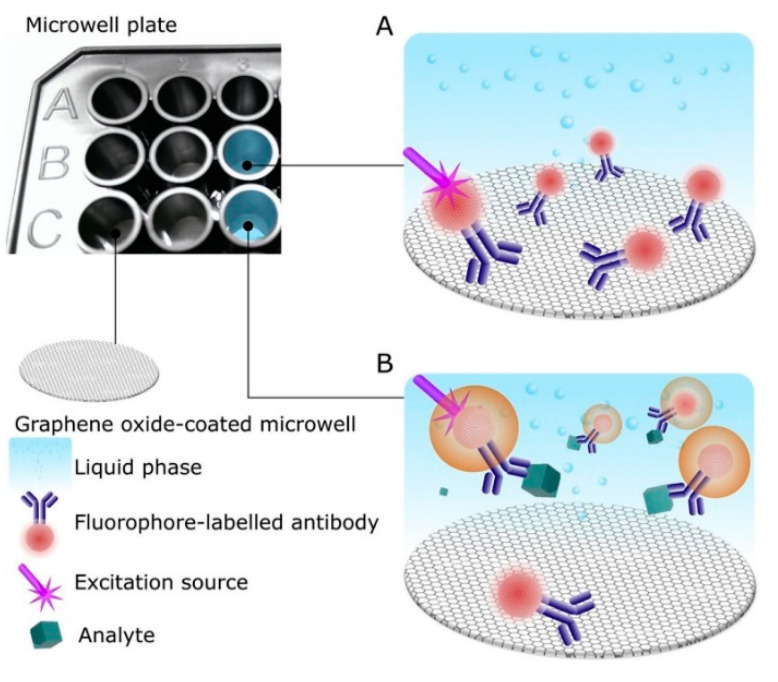
Optical immunosensor for prostate-specific antigen in urine. Fluorescent QDs-antibodies conjugates are quenched upon adsorption onto GO and immobilized onto a multi-well plate. Upon detection of the prostate-specific antigen, the resulting complex is formed and its affinity for GO decreases thereby increasing the fluorescence signal. Image adapted from [113] with permission of Elsevier, 2020.

**Figure 17 sensors-23-00949-f017:**
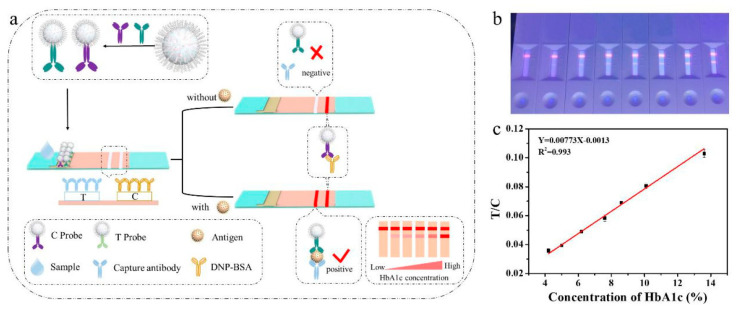
Optical immunosensor for glycosylated hemoglobin in blood samples. (**a**) Fluorescent BSA-wrapped CdSe-ZnSQDs conjugated to antibodies targeting the glycosylated hemoglobin are adsorbed onto a nitrocellulose membrane. (**b**,**c**) When the analyte is loaded, it binds to the fluorescent QDs that migrate towards a revelation zone where anti-human hemoglobin antibodies are present. A double fluorescent signal is detected; otherwise, in the absence of the analyte only the control fluorescent band is visible. Image adapted from [116] with permission of American Chemical Society, 2022.

**Figure 18 sensors-23-00949-f018:**
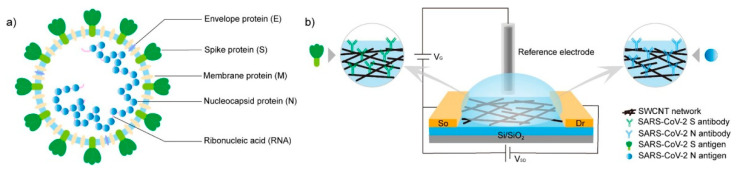
FET immunosensor for SARS-CoV-2 detection in nasopharyngeal samples. (**a**) Schematic structure of the SARS-CoV-2 viral particle. (**b**) Single-walled carbon NTs are functionalized with spike- or nucleoprotein-specific antibodies and placed on Au microchannels. Clinical samples can be applied onto the device to reveal both the spike and nucleoprotein antigens within 5 min through current and voltametric measurements. Image adapted from [122] with permission of American Chemical Society, 2021.

**Figure 19 sensors-23-00949-f019:**
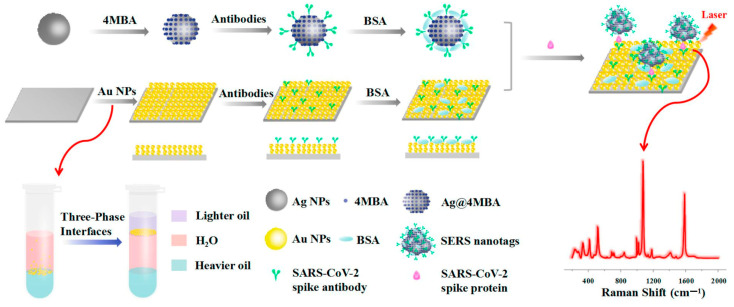
SERS-based immunosensor for SARS-CoV-2 detection in saliva. A two-layer AuNPs functionalized with antibodies on silicon wafers are sandwiched using AgNPs conjugated with secondary antibodies upon interaction with the viral particles. Upon detection of the analyte, the coupling between AgNPs and AuNPs increases the light scattered through plasmonic behavior. Image adapted from [124] with permission of Elsevier, 2021.

**Figure 20 sensors-23-00949-f020:**
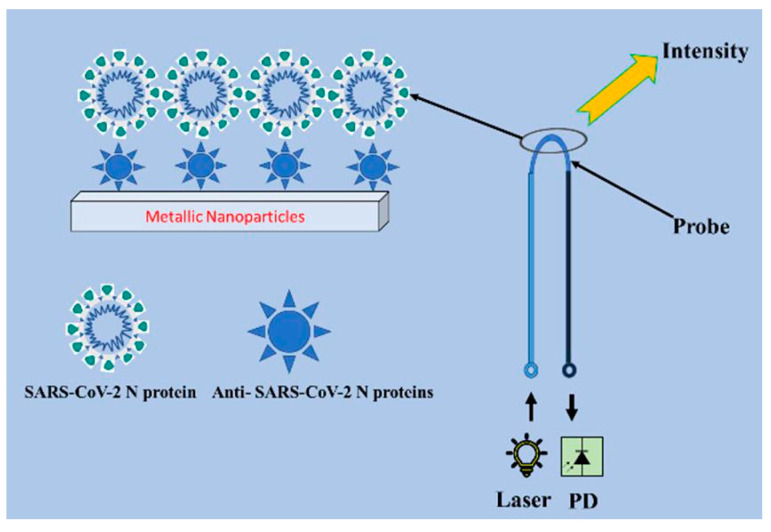
Immunosensor based on optical fiber for SARS-CoV-2 detection in oropharyngeal or nasopharyngeal swabs. Two-layer AuNPs structures can be immobilized onto the surface of a U-shaped plastic optical fiber and used to immobilize anti-nucleoprotein antibodies. The detection of viral particles triggers changes in the refracted light within the fiber that are enhanced by the plasmonic behavior of AuNPs. Image adapted from [125] with permission of Multidisciplinary Digital Publishing Institute, 2022.

**Figure 21 sensors-23-00949-f021:**
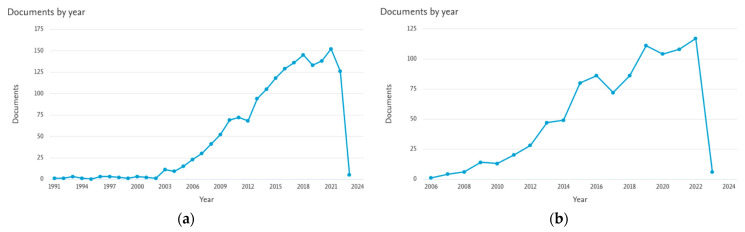
Trend of scientific papers concerning label-free immunosensors and aptasensors. (**a**) Survey of immunosensors according to the keywords “label free immunosensor” in Scopus database. (**b**) Survey of aptasensors according to the keywords “label free aptasensors” in Scopus database. Both surveys are obtained by searching within article title, abstract and keywords.

**Table 1 sensors-23-00949-t001:** Electrochemical aptasensors cited in this review.

Electrode	Main Components	Target	LOD; Range	Reference
Glassy carbon	Aptamers-functionalized AuNPs	Homocysteine in serum and urine	0.01 μM; 0.05–20.0 μM	[25]
Screen-printed carbon	Aptamers-functionalized flower-like Au microstructures	Serpin in plasma	0.031 ng mL^−1^; 0.039–10 ng mL^−1^	[26]
Screen-printed Au	Aptamers-functionalized CeO_2_NPs	Epithelial sodium channel in urine	0.012 ng mL^−1^; 0.05–3.0 ng mL^−1^	[27]
Glassy carbon	DNA nonotetrahedrons-functionalized AuNPs-IL-MoS_2_, aptamer-functionalized AuNPs-Fe-MIL-88 MOFs	Thrombin in serum	56 fM; 0.298–29.8 pM	[28]
Glassy carbon	Aptamers-functionalized reduced MoS_2_-AuNPs	Zearalenone and fumonisin B1 in food	5 × 10^−4^ ng mL^−1^; 1 × 10^−3^–10 ng mL^−1^ (zearalenone) and 1 × 10^−3^–1 × 10^2^ ng mL^−1^ (fumonisin B1)	[29]
Glassy carbon	Aptamers-functionalized (3-aminopropyl)triethoxysilane-modified GO-AgNPs	Chloramphenicol in food	3.3 pM; 10 pM–0.2 μM	[30]
Glassy carbon	Aptamers-functionalized thiourea capped-ZnSQDs-AuNPs	β-casomorphin 7 in urine	350 aM; 1 fM–0.6 μM	[31]
Glassy carbon	Aptamers-functionalized core-shell Cu-In-S-ZnSQDs-AgNPs-GQDs	Ractopamine in urine and serum	330 aM; 1 fM–901.4 nM	[32]
Magnetic glassy carbon	Aptamers-functionalized magnetic reduced GO-Fe_3_O_4_-Cu_2_O nanosheets and Ag-resorcinol-formaldehyde NPs-AgNDs	Prostate specific antigen in serum	6.2 pg mL^−1^; 0.01–100 ng mL^−1^	[33]
Glassy carbon	Aptamers-functionalized reduced GO-AgNPs and prussian blue-AuNPs	Acetamiprid in food	0.30 pM; 1 pM–1 μM	[34]
Glassy carbon	Aptamers-functionalized reduced GO-AuNPs	Glycated albumin	0.07 μg mL^−1^; 2–10 μg mL^−1^	[35]
Glassy carbon	Aptamers-functionalized MXene-AuNPs	Chloramphenicol in food	0.03 pM; 0.0001–10 nM	[36]
Glassy carbon	Aptamers-functionalized PtNPs-MIL-101(Fe) MOFs	Aflatoxin M1 in food	2 × 10^−3^ ng mL^−1^; 1 × 10^−2^–80 ng mL^−1^	[37]
Au	Aptamers-functionalized CoNi-MOFs	Enrofloxacin in food, water and serum	0.2 fg mL^−1^; 0.001–1 pg mL^−1^	[38]
Glassy carbon	Aptamers-functionalized AuNPs-Fe^3+^-catechol MOFs	Carcinoembryonic antigen in serum	0.33 fg mL^−1^; 1 fg mL^−1^–1 μg mL^−1^	[39]
Screen-printed carbon	Aptamers-functionalized carbon-black NPs-AuNPs	Cd(II) ions in water	0.14 ppb; 1–50 ppb	[40]
Screen-printed carbon	Aptamers-functionalized AuNPs-UiO-66-NH_2_ MOFs assisted by exonuclease	Streptomycin in food	2.6 pg mL^−1^; 0.005–150 ng mL^−1^	[41]
Fluorine-doped tin oxide	Aptamers-functionalized BiVO_4_-2D-C_3_N_4_ photoanode with II type heterojunction	Microcystin-LR in water	4.191 × 10^−8^ μg L^−1^; 5 × 10^−7^ μg L^−1^–10 μg L^−1^	[42]
Au	Aptamers-functionalized highly porous Au nanostructures	Acetamiprid in food	0.34 nmol L^−1^; 0.5–300 nmol L^−1^	[43]
Indium tin oxide	Aptamers-functionalized core-shell LaFeO_3_-g-C_3_N_4_ heterostructures	Streptomycin in food	0.0033 nM; 0.01–10000 nM	[44]
Au	Aptamers-functionalized GQDs-AuNPs composites	Vitamin D in serum	0.7 nM; 1–500 nM	[45]
Screen printed graphite	Aptamers-functionalized poly-3-amino-1,2,4-triazole-5-thiol-GO-AuNPs and secondary aptamer probes	Lipocalin-2 in blood serum	0.3 ng mL^−1^; 1–1000 ng mL^−1^	[46]
Indium tin oxide electrodes	Aptamers-functionalized TiO_2_-BiOI-BiOBr visible light photosensitive composite	Streptomycin food and blood serum	0.04 nM; 0.05–150 nM	[47]
Screen-printed carbon	Aptamers-functionalized magnetic GO-Fe_3_O_4_	Organophosphorus pesticides in food	N.D.	[48]
Pt	Aptamers-functionalized AgNPs and luminol-hydrogen peroxide	Kanamycin in food	0.06 ng mL^−1^; 0.5–100 ng mL^−1^	[49]
Au	Aptamers-functionalized polyethyleneimine-coated reduced GO-AuNCs and single-stranded DNA-binding protein	Chloramphenicol in food	2.08 pmol·L^−1^; 5 pmol·L^−1^–1 μmol·L^−1^	[50]
Au	Polyethyleneimine-graphite-like carbon nitride/AuNWs functionalized with SH-aptamers	Chloramphenicol in food	2.96 × 10^−9^ μM; 1 × 10^−7^–1 μM	[51]
Au	Aptamers-functionalized AuNSs conjugated to CdSQDs and PdSQDs	Kanamycin and tobramycin in serum	0.12 nM (kanamycin) and 0.49 nM (tobramycin); 1–4 × 10^2^ nM (kanamycin) and 1–1 × 10^4^ nM (tobramycin)	[52]
Au	DNA nanostructures and horseradish peroxidase-functionalized AuNPs assisted by exonuclease	Kanamycin in food	9.1 fg mL^−1^; 0.05 pg mL^−1^–10 ng mL^−1^	[53]
Carbon	Aptamer-coated Au nanodendrites and 4-mercaptophenylboric acid-thionine-functionalized AuNPs	Sialic acid in serum	60 nM; 0.1–440 μM	[54]

**Table 2 sensors-23-00949-t002:** Electrochemical immunosensors cited in this review.

Electrode	Main Components	Target	LOD; Range	Reference
Screen-printed carbon	AuNPs-antibodies	Epithelial sodium channel in urine	2.8 × 10^−1^ ng mL^−1^; 9.375 × 10^−2^–1.0 ng mL^−1^	[62]
Screen-printed carbon	CeO_2_NPs-antibodies	Herceptin-2 in serum	34.9 pg mL^−1^; 0.001–20.0 ng mL^−1^	[63]
Glassy carbon	Divalent metal ions-functionalized AuNPs-carbon NSs-antibodies	Prostate specific antigen, carcinoembryonic antigen and α-fetoprotein in serum	3.6 pg mL^−1^ (prostate specific antigen), 3.0 pg mL^−1^ (carcinoembryonic antigen) and 2.6 pg mL^−1^ (α-fetoprotein); 0.01–100 ng mL^−1^ (prostate specific antigen), 0.01–80 ng mL^−1^ (carcinoembryonic antigen and α-fetoprotein)	[64]
Au	Magnetic Fe_3_O_4_NPs-antibodies	Siglec 15 protein in serum	0.82 pg mL^−1^; 1 pg mL^−1^–100 ng mL^−1^	[65]
Screen-printed carbon	Nafion-embedded poly(2-hydroxyethyl methacrylate-glycidyl methacrylate)NPs-urease	Urea in serum	0.77 μM; 0.01–500 mM	[66]
Graphite	AgNPs-antibodies	Tick-borne encephalitis virus in serum	90 IU mL^−1^; 100–1600 IU mL^−1^	[67]
Glassy carbon	Antibodies-functionalized SnS_2_ nanoflakes-chitosan	Carcinoembryonic antigen in serum	5 pg mL^−1^; 0.006–3.0 ng mL^−1^	[68]
Screen-printed carbon	AgNPs-reduced GO-antibodies	Immunoglobulin G in serum	0.00086 ng mL^−1^; 0.001–0.05 and 0.05–50 ng mL^−1^	[69]
Screen-printed carbon	GO-antibodies	Epithelial sodium channels in urine	0.198 ng mL^−1^; 0.01–1.5 ng mL^−1^	[70]
Screen-printed	GO-Fe_3_O_4_NPs-Prussian blue-AuNPs-antibodies	Hepatitis B surface antibody in serum	0.166 pg mL^−1^; 0.5 pg mL^−1^–200 ng mL^−1^	[71]
Glassy carbon	Carboxyl G functionalized with mesoporous silica NPs-Methylene blue-AuNPs-antibodies	Galectin-3 in serum	2.0 fg mL^−1^; 50 fg mL^−1^–500 ng mL^−1^	[72]
Au	Carbon nanochips-embedded Au colloids-TiO_2_NPs-chitosan composites	β-lactoglobulin in food	0.01 pg mL^−1^; 0.01–500 pg mL^−1^	[73]
Screen-printed carbon	Chitosan-G-ionic liquid-ferrocene porous cryogel decorated with AuNPs-antibodies	Prostate-specific antigen in serum	4.8 × 10^−8^ ng mL^−1^; 1.0 × 10^−7^–1.0 × 10^−1^ ng mL^−1^	[74]
Indium tin oxide	Urease-functionalized ZnONPs-polyaniline-chitosan	Urea in serum	29.84 ppm; 20–500 ppm	[75]
Glassy carbon	Sandwich-like carbon NTs-CoS_2_-polyalanine-antibodies-horseradish peroxidase and AuNPs-antibodies	Carcinoembryonic antigen in serum	0.33 pg mL^−1^; 0.001–40 ng mL^−1^	[76]
Silicon oxide	Microfibers of single-walled carbon NTs-glucose oxidase	Glucose	N.D.	[77]
Glassy carbon	Vinyl ferrocene- and N-hydroxy succinimide acrylate-bifunctionalized carbon NTs conjugated with antibodies	α-fetoprotein in serum	1.14 ng·mL^−1^; 10 ng·mL^−1^–50 μg·mL^−1^	[78]
Glassy carbon	Carbon NTs-polyalanine functionalized with AuNPs-antibodies	Prostate-specific antigen in serum	0.5 pg·mL^−1^; 1.66 ag·mL^−1^–1.3 ng·mL^−1^	[79]
Titanium foil	Porous hydrogen titanate NTs-glucose oxidases	Glucose	59 μM; 1–10 mM	[80]
Glassy carbon	Antibodies-functionalzied sandwich-like AuNPs-embedded MOFs and MnO_2_ nanosheets-trimetallic Au-Pt-Pd nanocubes	Neuron-specific enolase in serum	4.17 fg mL^−1^; 10 fg mL^−1^–100 ng mL^−1^	[81]
Glassy carbon	Antibodies-functionalized bamboo-like sandwich-like carbon nanostructure-toluidine blue-functionalized copper-based MOFs	C-reactive protein in serum	166.7 pg mL^−1^; 0.5–200 ng mL^−1^	[82]
Glassy carbon	Sandwich-like antibodies-conjugated DNA dendrimers on antibodies-functionalized electrodes	Prostate-specific antigen in serum	0.26 pg mL^−1^; 1 pg mL^−1^–10 ng mL^−1^	[83]

**Table 3 sensors-23-00949-t003:** Optical aptasensors cited in this review.

Signal Detected	Main Components	Target	LOD; Range	Reference
Fluorescence	Aptamers-coated AuNPs, carbon QDs	Kanamycin in food	18 nM; 0.04–0.24 μM	[84]
Fluorescence	Aptamers-coated AuNPs, Rhodamine B	Carbendazim in water	2.33 nM; 2.33–800 nM	[85]
Fluorescence	Aptamers-coated AuNPs, Rhodamine B	Sulfamethazine in water and food	0.82 ng mL^−1^; 1.25–40 ng mL^−1^	[86]
Fluorescence	Aptamers-coated AuNPs, SYBR Green I-functionalized cDNA	Suplhadimetoxine in water and food	3.41 ng mL^−1^ (water) and 4.41 ng L^−1^ (food); 2 ng mL^−1^–300 ng mL^−1^	[87]
Fluorescence	Aptamers-functionalized AuNPs-carbon QDs	Adenosine triphosphate	20 μM: 20–280 μM	[88]
Fluorescence	Fluorescent aptamers-coated polydopamine NSs, DNase-I enzyme	Metalloproteinase-9 and -2 in urine and tissue homogenate	9.6 pg mL^−1^ (metalloproteinase-9) and 25.6 pg mL^−1^ (metalloproteinase-2); 24–600 pg mL^−1^ (metalloproteinase-9) and 64–1600 pg mL^−1^ (metalloproteinase-2)	[89]
Fluorescence	AuNRs coated with Ag nanoclusters-conjugated aptamers, exonuclease enzyme	Adenosine triphosphate in serum	26 pM; 50 pM–1.0 nM	[90]
Color	Polyethyleneimine-coated AuNPs, unfunctionalized aptamers	Chlorpyrifos in water and food	7.4 ng mL^−1^; 20–300 ng mL^−1^	[92]
Color	AuNPs coated with unfunctionalized polyA aptamers	Prostate specific antigen in serum	20 pg mL^−1^; 0.1–100 ng mL^−1^	[93]
Color	Unfunctionalized truncated aptamers-coated AuNPs	Bisphenol A in food and water	7.60 pM (38-mer) and 14.41 pM (12-mer); 20–100 pM	[94]
Color	Unfunctionalized aptamers-coated AuNPs	Chloramphenicol and tetracycline in food	32.9 nM (tetracycline) and 7.0 nM (chloramphenicol); 0.05–3.0 μM (tetracycline) and 0.05–1.8 μM (chloramphenicol)	[95]
Color	Unfunctionalized aptamers-coated AuNPs	*Bacillus carboniphilus* on biofilms	5 × 10^3^ CFU mL^−1^; 10^4^–10^7^ CFU mL^−1^	[96]
Color	Unfunctionalized RNA aptamers-coated AuNPs	Human papilloma-virus type 16 L1 in clinical and vaccine samples	9.6 ng mL-^1^; 9.6–201.6 ng mL^−1^	[97]
Color	Unfunctionalized aptamers-coated AuNPs	Acetamiprid in food and water	1.74 μM; 0–140 μM	[98]
Color	Glutamic acid-grafted cellulose acetate NFs functionalized with NH_2_-aptmaers, aptamer-complementary cDNA-coated AuNPs	Kanamycin in food and water	2.5 nM; 2.5–80 nM	[99]
Color	Truncated aptamers-coated AuNPs onto nitrocellulose membrane, aptamer-complementary cDNA	Oxytetracycline in food	5 ng mL^−1^; N.D.	[100]
Color	Unfunctionalized aptamers-coated onto nitrocellulose membrane	Gentamicin in food	300 nM; 10–1000 nM	[101]
Fluorescence	GO coated with fluoresceine-conjugated aptamers	Tropomyosin allergen in food	2 nM; 0–1.3 μM	[102]
Color	Enzyme-assisted, MnO_2_ nanosheets-induced AuNPs and biotin-labeled cDNA	Alkaline phosphate and ochratoxin A in food	0.05 U·L^−1^ (alkaline phosphate) and 5.0 nM (ochratoxin A); 6.25–750 nM (ochratoxin A)	[104]
Fluorescence	Fe_3_O_4_-GO-assisted AT-rich three-way junctions DNA-stabilized CuNPs	Isocarbophos in food and water	3.38 nM; 10–500 nM	[105]
Fluorescence	Fluorescent aptamers-coated zirconium-porphyrin MOFs	Chloramphenicol in food	0.08 pg mL^−1^; 0.1 pg mL^−1^–10 ng mL^−1^	[106]

**Table 4 sensors-23-00949-t004:** Optical immunosensors cited in this review.

Signal Detected	Main Components	Target	LOD; Range	Reference
Fluorescence	AuNPs–antibodies conjugates, secondary fluorescent antibodies	Epithelial sodium channel in blood	N.D.	[107]
Color	Magnetic Au-coated Fe_3_O_4_NPs–antibodies conjugates	Glyphosate in tap water	20 ng∙L^−1^; 0.01–100 μg∙L^−1^	[109]
Color	AuNPs–antibodies and Fe_3_O_4_NPs–antibodies conjugates	Prostate specific antigen in serum	0.009 ng mL^−1^; 0.01–20 ng mL^−1^	[110]
Fluorescence	Fluorescent AuNPs–albumin, -rhodamine B and -β-lactoglobulin conjugates	Nephrin and podocin in urine	N.D.	[111]
SPR	Magnetic AuNPs-antibodies, SPR active sensor disks	CD5 in serum	8.31 fM; N.D.	[112]
Fluorescence	Fluorescent QDs-antibodies, GO-coated multi-well plates	Prostate specific antigen in urine	0.05 ng mL^−1^; 0.15–10 ng mL^−1^	[113]
Fluorescence	Antibodies-conjugated CdSe-ZnSQDs and AuNRs	Porcine reproductive and respiratory syndrome virus in swine serum	0.55 TCID_50_ mL^−1^; 10^1^–3.5 × 10^4^ TCID_50_ mL^−1^	[114]
Fluorescence	Rhodamine-labelled peptides adsorbed onto WS_2_-Pt-Fe_2_O_3_ micromotors	Bacterial lipopolysaccharides	0.12 nM; 4.0–1,000,000 ng mL^−1^	[115]
Fluorescence	Fluorescent bovine serum album-wrapped CdSe-ZnSQDs conjugated to antibodies targeting the glycosylated hemoglobin, nitrocellulose membrane	Glycosylated hemoglobin in blood	N.D.	[116]
SERS	Antibodies-conjugated AuNSs, nitrocellulose membrane	Metalloproteinase-9, S100 calcium-binding protein B and neuro-specific enolase in blood	0.01 pg mL^−1^; 0.0001–1000 ng mL^−1^	[117]
Absorbance	Antibodies-conjugated magnetic Fe_3_O_4_ NPs coated with a gold shell	Human growth hormone in serum	0.082 nmol L^−1^; 0.1–5.0 nmol L^−1^	[118]
Reflected light	Analyte-induced gold deposition on antibodies-conjugated Mn-ZnS QDs, microscope coverslip	Prostate-specific antigen in serum	3.5 × 10^−3^ pg mL^−1^; 0.01–100 pg·mL^−1^	[119]

## Data Availability

Not applicable.

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
