# Peer review of "Bio-Tailored Sensing at the Nanoscale: Biochemical Aspects and Applications"

_sensors, 2023, doi:10.3390/s23020949_

Round 1

Reviewer 1 Report

The manuscript written by Fata et al gives a minireview of the recent progress on label-free electrochemical biosensors and aptasensors. The manuscript is carefully written and the logic is clear, so I recommend a minor revision for publishing in Sensors with the following issues being addressed:

1.        The word “bio-tailored” in the title “Bio-tailored sensing at the nanoscale: biochemical aspects and Applications” lacks interpretations in this manuscript.

2.        In lines 114-115, the authors claimed “Aptamers are short, single-stranded nucleotide sequences that spontaneously form hairpin-like (stem loop) structures in solution” which is not completely correct. First, Aptamers could be either oligonucleotide or peptide biomolecules. Secondly, the secondary structures of the aptamers are diverse, such as hairpin, stem-loop, pseudoknot, or G-quartet etcetera.

3.        Eliminate the meaningless words to make the manuscript straightforward. For example, in Line 936  “It is not surprising that” could be deleted.

Author Response

We thank the reviewer for his comments. 

A point-by-point response is reported herein:

  1. The meaning of "bio-tailored" has been clarified in the Abstract (page 1) and Introduction (page 2)
  2. We agree. The sentence has been re-arranged to provide a more general description of aptamers
  3. Meaningless and redundant words and sentences have been deleted throughout the text when appropriate

Reviewer 2 Report

The present review "Bio-tailored sensing at the nanoscale: biochemical aspects and applications" report the development of electrochemical and optical biosensor (aptasensors) over the last 2 years. The review is well-structured and written straightforwardly. The authors divided the text into sections, where the first is dedicated to electrochemical biosensors, the second to optical ones, the third section is dedicated to Covid research, and the last section is a brief discussion about the conclusions and outlook. Each section is written with a good discussion about the works cited. Below one can find a few comments.

1) In the introduction, the authors could provide one schematic representation of the discussed topics, for example, label-free. It can be clear for those from the area but adding a figure to support the text could be an attractive broad spectrum of researchers.

2) In the following discussion, some pieces of information were missing (especially during the electrochemical section), and the authors could provide this information to improve the text. Regarding “tailored” sensors, details about: the mechanism of response, incubation time (are only cited on page 28-line 832), and BSA application for blocking unreacted surfaces, were not highlighted, and they play a crucial role in the sensor development and strategy. The interaction between the receptor and the target is well-discussed, however, the response mechanism is not clear.

For example, using the Fe(CN)6, one can use the “gate-effect” strategy or can attach it to the surface where the interaction with the analyte decrease or increase the charge transfer, resulting in the current signal changes. Also, one can use oxygen as a redox probe instead the metallic complexes.

3) In the optical biosensors, the representative images are better where most of them are showing the response mechanism.

Author Response

We thank the reviewer for his comments.

A point-by-point response is reported herein:

  1. A new figure has been made and placed within the Introduction (Figure 1, page 3). The figure is intended to provide an overview of the main nanometric components used to build the hybrid sensors and the electrochemical or optical detection systems used
  2. A short discussion about the mechanism of response, the use of blocking agent to avoid background noise and the importance of the incubation time related to the electrochemical sensors has been added (pages 4 and 10). Though we agree with the reviewer, we intentionally decided to not stress these aspects as our manuscript was focused on reporting the main biomolecules and nanomaterials used to build hybrid devices rather than exhaustively describe more technical aspects
  3. We agree with the reviewer.